# iFedDR: Auto-Tuning Local Computation with Inexact Douglas-Rachford Splitting in Federated Learning

## Abstract

Federated learning usually requires specifying the amount of local computation needed a priori. In this work, we instead propose a systematic scheme to automatically adjust and potentially reduce the local computations while preserving convergence guarantees. We focus on proximal-based methods, where we demonstrate that the proximal operator can be evaluated inexactly up to a relative error, rather than relying on a predefined sequence of vanishing errors. Our proposed method, iFedDR, is based on a novel error-corrected version of inexact Douglas-Rachford splitting. It mitigates the need for hyperparameter tuning the number of client steps, by triggering refinement on-demand. We derive iFedDR as an instance of a much more general construction, which allows us to handle minimax problem, and which is interesting in its own right. Several numerical experiments are carried out demonstrating the favorable convergence properties of iFedDR.

## 1 Introduction

Federated learning (FL) has emerged as a promising approach to distributed machine learning that allows multiple clients to collaboratively train a model. In an attempt to keep communication costs low, the clients repeatedly apply their update rule $\tau$ times before centrally aggregating the model and proceeding to the next round.

A popular instance of this general idea is FedAvg (McMahan et al., 2017) which locally takes $\tau$ gradient descent steps. Unfortunately, FedAvg requires a very small client stepsize in data heterogeneous settings to avoid the so called client-drift. This requirement appears theoretically as a horizon dependent $O(1/\tau)$ client stepsize, which intuitively is used to prevent the $\tau$ steps to move beyond a single (large) step of gradient descent (see e.g. Karimireddy et al. (2020, Thm. I)). Because of the small stepsize the benefit of multiple local updates is unclear under data heterogeneous.

A more principled approach is arguably to instead use the local computation to approximate a well-understood update rule such as the proximal operator, which is known to be more stable and usually allows for larger stepsizes. FedProx (Li et al., 2020) was the first to use proximal operators for FL and did so by directly averaging the client updates. This straightforward application of the prox turns out to not necessarily converge to a solution when using fixed stepsize (Pathak & Wainwright, 2020, Prop. 2) and proper convergence relies on behaving like FedAvg with $\tau = 1$ by letting the proximal stepsize $\gamma$ go to zero (Malekmohammadi et al., 2022, Thm. 1). In other words, it might be argued that one should simply use FedAvg with $\tau = 1$ instead.

Fortunately there is a "correct" way to employ the proximal operator such that we can benefit from its strong properties mentioned previously. By leveraging the operator splitting literature, FedSplit (Pathak & Wainwright, 2020), FedDR (Tran Dinh et al., 2021) and FedPi (Malekmohammadi et al., 2022) fixes the update rule of FedProx. However, all of these prox-based method still need the approximation errors of the proximal operator to be absolutely summable in the sense that the positive approximation error sequence $(\varepsilon_k)_{k \in \mathbb{N}}$ satisfies $\sum_{k=1}^{\infty} \varepsilon_k < \infty$, but no further guidance is provided as how to select them. This translates into requiring an increasing number of local updates that problematically needs to be specified preemptively.

This paper specifically looks at the approximation quality of the clients proximal update and ask:

*Can we automate the number of local updates needed?*

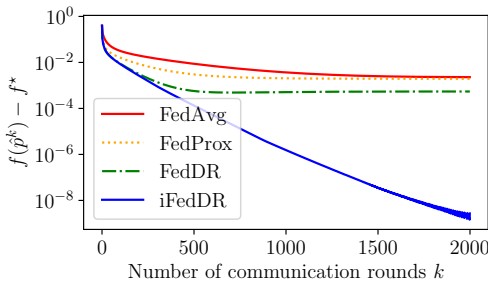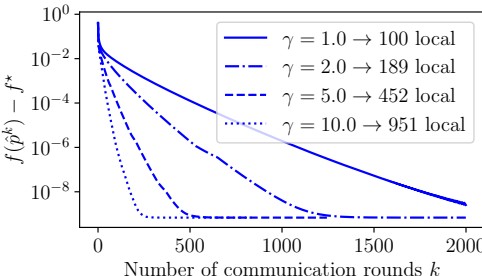

Figure 1: Logistic regression on the heterogeneous vehicle dataset. (left) Similar to FedAvg and Fed-Prox, FedDR does not necessarily converge if the proximal approximation is not accurate enough, whereas iFedDR can converge by automatically adjusting the approximation quality to a sufficient level. (right) The proximal stepsize $\gamma$ can be taken large to speed up convergence and iFedDR will automatically increase the number of local steps accordingly (cf. Remark 3.2 regarding the tradeoff).

We answer this in the affirmative by making the following contributions:

- We introduce the error-corrected algorithm, iFedDR, which automatically checks if refining the approximate evaluation of the proximal mapping is necessary through a computationally negligible error condition. This automatic adjustment mitigates the need for pre-determining an appropriate number of client iterations, while allowing for large stepsizes (see Figure 1). The construction is made possible by introducing a *relative* error condition, which enables the error condition at each communication round to be treated separately.

- Our analysis for iFedDR extends to monotone inclusion problems and consequently enjoys provable guarantees not only for (constrained) convex minimization but also for the important case of (constrained) minimax problems, and more generally $m$-player games. We obtain iFedDR through a novel inexact preconditioned proximal point algorithm (iPPPA) involving a semidefinite preconditioner and a correction step mitigating the effect of the inexactness, which might be of independent interest.

- We demonstrate the favorable properties of iFedDR under heterogeneous data distributions. A simple rule is proposed for minimizing the number of refinement calls. We observe that iFedDR either match or improve the performance of the baselines without needing tuning.

### 1.1 Comparison with existing methods for federated learning

The issue of client drift for FedAvg is made precise in the (tight) lower bound $\Omega(H^{2/3}/m^{2/3})$ for convex and smooth functions in Glasgow et al. (2022, Thm. 3.3), where $m$ is the number of outer iterations and $H$ is the heterogeneity constant in $\|\nabla f_i(x)\|^2 \leq H^2 + 2LB^2(f(x) - f^\star)$. The bound says that FedAvg *cannot* avoid depending on the level of heterogeneity $H$ and that the error decreases relatively slowly as $1/m^{2/3}$.

To remove the dependency on the heterogeneity constant $H$ and eventually obtain a $O(1/m)$ rate, FedDR (Tran Dinh et al., 2021) uses the local computation to instead approximate a proximal operator on the clients, which are then integrated based on Douglas-Rachford splitting. The proposed iFedDR method is an extension, that incorporates an adaptive stepsize and an extragradient correction step, to allow for the relaxed condition of relative inexactness. The relative error condition mitigates the need for hyperparameter tuning of the number of client iterations. The iFedDR scheme additionally enjoys provable guarantees for minimax problems. In contrast, the analysis technique used in (Tran Dinh et al., 2021) relies on the Douglas-Rachford envelope (Themelis & Patrinos, 2020) and as such cannot be extended to the minimax setting. The (only) point of contact between iFedDR and FedDR is for minimization problems, when the proximal operators can be computed exactly, which is very rarely the case. See Appendix E for further details on comparison with FedDR.

FedDR itself can be seen as a relaxed version of FedSplit (Pathak & Wainwright, 2020) (FedSplit can be recovered from FedDR by taking $\lambda_k = 2$). We note that without the relaxation it is not possible to

Table 1: Comparison of iFedDR with existing algorithms under convexity and $L_i$-smoothness of the client objectives after $m$ global iterations using $\tau$ local steps. The parameter $H$ is the heterogeneity constant. Due to the error-correction step, the proposed iFedDR method achieves a $O(\frac{1}{\gamma^3 m})$ communication rate, while automatically setting the number of inner steps. The stepsize $\gamma$ can be taken large at the expense of a harder client subproblem. We use $\widetilde{O}$ to hide logarithmic factors.

| Method | Minimax | Adaptive $\tau$ | Comm. rate | Local cond. number | Proximal stepsize $\gamma$ | Reference |
|--------|---------|-----------------|------------|--------------------|-----------------------------|-----------|
| GD | ✗ | ✗ | $O(\frac{L}{m})$ | - | - | (Nesterov et al., 2018) |
| FedAvg | ✗ | ✗ | $\Omega(\frac{L}{\tau m} + \frac{L^{1/3}H^{2/3}}{m^{2/3}})^2$ | - | - | (McMahan et al., 2017) |
| Scaffold | ✗ | ✗ | $\widetilde{O}(\frac{L}{m})$ | - | - | (Karimireddy et al., 2020, Thm. 3) |
| FedProx | ✗ | ✗[1] | $O(\frac{1}{\sqrt{m}})^3$ | $\kappa_i = 1 + \gamma L_i$ | $\gamma \to 0$ | (Li et al., 2020) |
| FedDR | ✗ | ✗[1] | $O(\frac{1}{m\gamma^3})$ | $\kappa_i = 1 + \gamma L_i$ | $\gamma \in (0, 1/L)$ | (Tran Dinh et al., 2021, Thm. 3.1)[4] |
| iFedDR | ✓ | ✓ | $O(\frac{1}{m\gamma^3})$ | $\kappa_i = 1 + \gamma L_i$ | $\gamma > 0$ | Theorem 3.1 |

[1] FedProx/FedDR requires a predefined summable error sequence regarding the client proximal updates.
[2] The lowerbound for FedAvg is due to Glasgow et al. (2022, Thm. 3.3).
[3] Convergence of FedProx requires decreasing stepsize sequence $\gamma_k$ (Malekmohammadi et al., 2022, Thm. 1).
[4] FedDR is analysed under nonconvex and Lipschitz continuous client objectives.

show convergence for convex problems without modifying the objective. For a detailed comparison see Ryu & Yin (2022, Sec. 2.7.1) and Malekmohammadi et al. (2022, Sec. 3.3-3.4).

Another approach to removing the dependency on the heterogeneity constant $H$, is the Scaffold algorithm, which uses the variance reduction technique SAGA to eventually obtain a rate of $\widetilde{O}(1/m)$ in Karimireddy et al. (2020, Thm. III). The iFedDR algorithm uses the same level of communication per iteration as Scaffold while saving on the memory usage on the server. In contrast with Scaffold, iFedDR also extends to minimax problems and constrained problems with provable guarantees.

In the special case of strongly convex and smooth objectives, an acceleration of the communication complexity was shown through ProxSkip (Mishchenko et al., 2022) by taking an (expected) number of client steps with dependency on the conditioning number. The dependency was later improved in followup works (Maranjyan et al., 2022; Sadiev et al., 2022). In contrast with these works, iFedDR applies to general convex problems and as a byproduct does not require knowledge of the strong convexity modulus. Our work takes an orthogonal direction to reducing the overall computation by avoiding hyperparameter tuning of the client steps. Note that the prox referred to in ProxSkip is used to capture the consensus constraint and is thus distinct from the one which is computed inexactly in iFedDR.

An overview of the comparison is provided in Table 1 of Appendix A.

## 2 Problem setup & preliminaries

In order to capture both minimization problems and minimax problems we will be interesting in the following finite sum inclusion problem, which seeks to find $x \in \mathbb{R}^n$ such that

$$0 \in \frac{1}{N} \sum_{i=1}^{N} F_i(x) + G(x), \tag{2.1}$$

where each $i \in [N]$ should be understood as a separate client, each only having access to their associated operator $F_i : \mathbb{R}^n \to \mathbb{R}^n$, while the server has access to the possible set-valued mapping $G : \mathbb{R}^n \rightrightarrows \mathbb{R}^n$. We will make the following assumptions (see Appendix B for any missing definitions).

**Assumption I** (Requirements for problem (2.1))**.**

(i) *The operators $F_i : \mathbb{R}^n \to \mathbb{R}^n$ for all $i \in [N]$ and $G : \mathbb{R}^n \rightrightarrows \mathbb{R}^n$ are maximally monotone.*

(ii) *The operator $F_i$ is $L_i$-Lipschitz continuous for all $i \in [N]$, i.e.*

$$\|F_i(x) - F_i(x')\| \le L_i \|x - x'\| \quad \forall x, x' \in \mathbb{R}^n.$$

Maximal monotonicity allows us to capture a large range of problems including (possibly constrained) convex minimization problems, convex-concave minimax problems and more generally convex $m$-player games. We provide two prominent examples that can be cast as the inclusion (2.1) below.

**Example 2.1** (Minimization). Consider the following convex optimization problem

$$\text{minimize}_{x\in\mathbb{R}^n} \ \frac{1}{N}\sum_{i=1}^N f_i(x) + g(x) \tag{2.2}$$

The first order stationary condition may be written in the form of the inclusion (2.1) by setting $F_i = \nabla f_i$ and $G = \partial g$. Maximally monotonicity of the operators are satisfied when $g$ is proper lsc convex and $(f_i)_{i\in[N]}$ are convex and smooth (cf. Appendix D.3 for details). □

**Example 2.2** (Minimax). Consider the following optimization problem

$$\text{minimize}_{u\in\mathbb{R}^m}\text{maximize}_{v\in\mathbb{R}^r} \ g(u) + \frac{1}{N}\sum_{i=1}^N f_i(u,v) - h(v) \tag{2.3}$$

Denoting $x = (u,v) \in \mathbb{R}^n$ where $n = m + r$, the first order stationary condition may be written in the form of the inclusion (2.1) with $G(x) = (\partial g(u), \partial h(v))$, $F_i(x) := (\nabla_u f_i(u,v), -\nabla_v f_i(u,v))$. Maximally monotonicity of the operators are satisfied when $g, h$ are proper lsc convex and $(f_i)_{i\in[N]}$ are convex-concave and Lipschitz continuous (cf. Appendix D.4 for details). □

As the operators $F_i$ are not accessible centrally, let $\boldsymbol{x} := (x_1, ..., x_N) \in \mathbb{R}^{Nn}$ and consider an equivalent but lifted consensus formulation of (2.1) that we shall refer to as the *primal* inclusion in the lifted space

$$0 \in T_{\text{P}}(\boldsymbol{x}) := A(\boldsymbol{x}) + B(\boldsymbol{x}) \tag{2.4a}$$

with

$$A(\boldsymbol{x}) := \frac{1}{N}(F_1(x_1), \dots, F_N(x_N)), \quad B(\boldsymbol{x}) := N_C(\boldsymbol{x}) + \frac{1}{N}(G(x_1), \dots, G(x_N)) \tag{2.4b}$$

where $C := \{\boldsymbol{x} \in \mathbb{R}^{Nn} \mid x_1 = x_2 = \dots = x_N\}$ is the consensus set, and $N_C = \partial\delta_C$ denotes the normal cone operator of the set $C$, while $\delta_C$ is the indicator function defined as $\delta_C(\boldsymbol{x}) = 0$ if $\boldsymbol{x} \in C$ and $+\infty$ otherwise. Maximal monotonicity of the operators $A$, $B$ follows from that of $F_i, G$, and $N_C$, and since maximal monotonicity is closed under addition (Bauschke & Combettes, 2017, Prop. 20.23, Ex. 20.26, Cor. 25.5). The dual problem associated with (2.4) consists of finding $\boldsymbol{y} \in \mathbb{R}^{Nn}$ such that

$$0 \in T_{\text{D}}(\boldsymbol{y}) := -A^{-1}(-\boldsymbol{y}) + B^{-1}(\boldsymbol{y}). \tag{2.5}$$

The primal dual problem then consists of finding $\boldsymbol{z} = (\boldsymbol{x}, \boldsymbol{y}) \in \mathbb{R}^{2Nn}$ such that

$$0 \in T_{\text{PD}}(\boldsymbol{z}) := \begin{bmatrix} A(\boldsymbol{x}) \\ B^{-1}(\boldsymbol{y}) \end{bmatrix} + \begin{bmatrix} \boldsymbol{y} \\ -\boldsymbol{x} \end{bmatrix}. \tag{2.6}$$

The operator $T_{\text{PD}}$ is maximally monotone owing to the fact that skew symmetric linear operators are maximally monotone, and since maximal monotonicity is closed under the inverse and addition operators (Bauschke & Combettes, 2017, Prop. 20.22, 20.23, Cor. 25.5). A point $\boldsymbol{z}^\star = (\boldsymbol{x}^\star, \boldsymbol{y}^\star)$ solves the primal dual problem if and only if $\boldsymbol{x}^\star$ is a solution to the primal problem (2.4) and $\boldsymbol{y}^\star$ solves the dual problem (2.5).

When the resolvents of the operators $A$ and $B$ are available, a classical approach to solving both the primal problem (2.4a) and the dual problem (2.5) is the Douglas-Rachford splitting (DRS) algorithm (Douglas & Rachford, 1956):

$$\boldsymbol{u}^k = (\text{id} + \gamma A)^{-1}(\boldsymbol{s}^k)$$
$$\boldsymbol{v}^k = (\text{id} + \gamma B)^{-1}(2\boldsymbol{u}^k - \boldsymbol{s}^k) \tag{DRS}$$
$$\boldsymbol{s}^{k+1} = \boldsymbol{s}^k + (\boldsymbol{v}^k - \boldsymbol{u}^k)$$

with a stepsize $\gamma > 0$. The iterates $(\boldsymbol{u}^k)_{k\in\mathbb{N}}$ and $(\boldsymbol{v}^k)_{k\in\mathbb{N}}$ are guaranteed to converge to a solution of the primal problem (2.4a), while $(\boldsymbol{y}^k)_{k\in\mathbb{N}} = (\gamma^{-1}(\boldsymbol{u}^k - \boldsymbol{s}^k))_{k\in\mathbb{N}}$ converges to a solution of the dual problem (2.5), when $A, B$ are maximally monotone and a primal solution exists (Eckstein & Bertsekas, 1992). An alternative to (DRS) for solving (2.4a) is the forward-backward splitting (FBS), $(\text{id} + \gamma B)^{-1}(\text{id} - \gamma A)$. However, FBS may not converge when the operator $A$ is merely monotone and the condition on the stepsize $\gamma$ is more stringent. In the next section we present iFedDR which we obtain as an instance of an inexact and relaxed variant of DRS, that incorporates an error-correction step enabling a relative error condition.

## 3 THE iFedDR ALGORITHM

We solve the primal dual problem (2.6) (and thus the original finite sum problem (2.1) through the consensus reformulation (2.4)) with an error-corrected version of an inexact Douglas-Rachford splitting (DRS) scheme that we refer to as iFedDR (Algorithm I). The update may appear difficult to arrive at directly, and we indeed find it by casting DRS as a *preconditioned* proximal update

---

**Algorithm I** The inexact federated Douglas-Rachford algorithm for problem 2.1 (iFedDR)

---

REQUIRE    starting point $s_i^{-1} \in \mathbb{R}^n$, $\alpha_{-1} = 0 \in \mathbb{R}$, stepsize $\gamma \in (0, \infty)$, $\lambda \in (0, 2)$, and $\sigma \in (0, 1)$

REPEAT FOR $k = 0, 1, \ldots$ until convergence

I.1: Each client $i \in [N] := \{1, \ldots, N\}$ computes
$$s_i^k = s_i^{k-1} - \lambda \alpha_{k-1}(\bar{x}_i^{k-1} - \hat{p}^{k-1})$$
and approximately compute the resolvent
$$\bar{x}_i^k \simeq (\text{id} + \gamma F_i)^{-1}(s_i^k) \tag{3.3}$$
and sends to the server
$$(\bar{x}_i^k, F_i(\bar{x}_i^k), s_i^k).$$

I.2: The server computes the error-corrected average $\hat{p}^k = (\text{id} + \gamma G)^{-1}(\frac{1}{N} \sum_{i=1}^N (\bar{x}_i^k - \gamma F_i(\bar{x}_i^k)))$ and the scalar quantities
$$\xi_k = \sum_{i=1}^N \|\bar{x}_i^k - \hat{p}^k\|^2, \quad \zeta_k = \frac{1}{\gamma^2} \sum_{i=1}^N \|\gamma F_i(\bar{x}_i^k) - s_i^k + \hat{p}^k\|^2, \quad \text{and} \quad \mu_k = \sum_{i=1}^N \langle \bar{x}_i^k - \hat{p}^k, s_i^k - \gamma F_i(\bar{x}_i^k) - \hat{p}^k \rangle. \tag{3.4}$$

See Remark D.3 for how to carry out the computation memory-efficiently.

I.3: IF $\sum_{i=1}^N \|s_i^k - \gamma F_i(\bar{x}_i^k) - \bar{x}_i^k\|^2 \le \sigma^2 \max\{\xi_k, \zeta_k\}$ THEN
the server sends back
$$(\hat{p}^k, \alpha_k) \quad \text{where} \quad \alpha_k = \mu_k/\xi_k.$$

I.4: ELSE
request the clients to refine the approximation in (3.3) to higher accuracy.

RETURN $\hat{p}^k$

---

and instead working on this simpler abstraction level as detailed in Sections 4 and 5. The algorithm includes an adaptive stepsize $\alpha_k$, an extragradient error-correction step, and a relative error condition which can be efficiently computed on the server as commented on in Remark D.3.

**Client subproblem**    Approximately computing the resolvent in (3.3) on the client amount to finding an approximate zero to the following strongly monotone operator under Assumption I(i)
$$F_i^\gamma(x) := F_i(x) + \frac{1}{\gamma}(x - s_i^k), \tag{3.1}$$
with the stepsize parameter $\gamma > 0$ (see Appendix D.3 for the special case of minimization). Due to having an explicit optimization problem on the client, the approach becomes modular: The subproblem may be solved in a variety of ways, e.g. with first-order methods such as the gradient method or the extragradient method (Korpelevich, 1976). For minimization problem, where $F_i = \nabla f_i$ reduces to a gradient of some cost function $f_i$, one may employ gradient descent, or its accelerated variants; in case of additional structure such as scenarios where the client objectives $f_i$ itself is a finite sum, variance reduction techniques may be employed.

**Progress measure**    As a measure of progress we track the *natural residual* in the lifted space defined as follows
$$\mathcal{G}_\gamma(\boldsymbol{x}) := \frac{1}{\gamma}(\boldsymbol{x} - (\text{id} + \gamma B)^{-1}(\boldsymbol{x} - \gamma A \boldsymbol{x})) \tag{3.2}$$
It is immediate that $\mathcal{G}_\gamma(\boldsymbol{x}) = 0$ if and only if $\boldsymbol{x} = (x, x, \ldots, x) \in \mathbb{R}^{Nn}$ and $x$ is a solution to the primal problem (2.4), i.e., that $x$ is a solution to the original problem (2.1).

**Theorem 3.1.** *Suppose the operators in* (2.4) *satisfy Assumption I and* $\text{zer}(A + B) \ne \emptyset$. *Let* $(\bar{\boldsymbol{x}}^k := \frac{1}{N}(\bar{x}_1^k, \ldots, \bar{x}_N^k))_{k \in \mathbb{N}}$ *be generated by iFedDR (Algorithm I) and* $\boldsymbol{s}^0 = \frac{1}{N}(s_1^0, \ldots, s_N^0)$. *Then,*

*(i) The iterates* $(\bar{\boldsymbol{x}}^k)_{k \in \mathbb{N}}$ *converges to some* $\boldsymbol{x}^\star \in \text{zer}(A + B)$.

*(ii) For all* $\boldsymbol{s}^\star = \boldsymbol{x}^\star + \gamma A \boldsymbol{x}^\star$ *where* $\boldsymbol{x}^\star \in \text{zer}(A + B)$, *we have that*
$$\min_{k=0,1,\ldots,m} \|\mathcal{G}_\gamma(\bar{\boldsymbol{x}}^k)\|^2 \le \frac{\|\boldsymbol{s}^0 - \boldsymbol{s}^\star\|^2}{\tau(m+1)}$$
*where* $\tau = \frac{(2-\lambda)\lambda\gamma(1+\gamma^2)}{(1-\sigma)^2}$ *and $m$ is the total number of communication rounds.*

**Remark 3.2.** Theorem 3.1 is obtained as a corollary of the more general Theorem 5.1 concerning iPPPA, which is covered in Sections 4 and 5 (see Appendices D.1 and D.2 for the reduction). Some remarks are in place. *(i) Communication complexity:* Theorem 3.1 implies, that to guarantee an $\epsilon$-accurate solution, iFedDR only needs $m = O(\frac{1}{\gamma(1+\gamma^2)\epsilon})$ communication rounds. *(ii) Heterogeneity agnostic:* The theorem requires no heterogeneity assumption, i.e. the client operators $F_i : \mathbb{R}^n \to \mathbb{R}^n$ can be arbitrarily different. *(iii) Linear speedup in the number of clients:* A centralized variant would naively need access to $\sum_{i=1}^N F_i(z)$, whereas this computation can be parallelized amongst $N$ clients with iFedDR, while still maintaining a $O(1/m)$ rate. *(iv) Accuracy-agnostic local solvers:* The errors from the proximal client solvers do not propagate to the rate in Theorem 3.1 due to the error-correction step in iFedDR. Thus, the number of client updates does not have to be specified preemptively based on the desired accuracy of the solution. *(v) A natural tradeoff:* A larger proximal stepsize $\gamma$ leads to a lower communication complexity through the factor $\frac{1}{\gamma(1+\gamma^2)}$. The price to pay is a more expensive proximal subproblem on the clients, since the conditioning number of the strongly monotone and Lipschitz continuous subproblem, $\kappa_i = 1 + L_i\gamma$, becomes larger with increasing $\gamma$ (cf. Figure 1). $\qquad\square$

**Linear convergence**   When dealing with minimization problems, linear convergence rate can be established for the class of *piecewise linear-quadratic* (PLQ) functions (Rockafellar & Wets, 2011).

**Definition 3.3** (Piecewise linear-quadratic). *A function $\varphi : \mathbb{R}^n \to \overline{\mathbb{R}}$ is called piecewise linear-quadratic (PLQ) if its domain can be represented as the union of finitely many polyhedral sets, and in each such set it is given by an expression of the form $\frac{1}{2}\langle x, Hx \rangle + \langle d, x \rangle + c$, for some $c \in \mathbb{R}$, $d \in \mathbb{R}^n$, and symmetric matrix $H \in \mathbb{R}^{n \times n}$.*

Piecewise linear-quadratic functions are widespread in application and include affine functions, quadratic forms, indicators of polyhedral sets, polyhedral norms such as the $\ell_1$-norm, regularizers such as elastic net, hinge loss, and many more, see (Aravkin et al., 2013).

We proceed to establish linear convergence for iFedDR when taking $F_i = \nabla f_i$ and $G = \partial g$ in (2.1) with $f_i$ and $g$ being PLQ. Notice that this assumption does not imply that the set of solutions is a singleton. Its proof is similar to (Latafat et al., 2019, Lem. IV), and relies on the fact that subdifferential of $f_i$ and $g$ as well as the normal cone of the consensus set are polyhedral sets implying that the operator $T$ defined in (4.2) is metrically subregular at all points in its graph. Consequently, the following corollary of Theorem 5.4 shows that iFedDR achieves linear rate for such problem classes.

**Corollary 3.4** (Piecewise linear-quadratic functions). *Suppose that the functions $f_i$ and $g$ are piecewise linear-quadratic. Let $F_i = \nabla f_i$ and $G = \partial g$ in (2.1). Then, the sequences $(s_i^k)_{k \in \mathbb{N}}$ generated by iFedDR (Algorithm I) converge R-linearly to zero.*

## 4   DRS through inexact proximal point algorithm

To arrive at iFedDR (Algorithm I) we will start by considering a general inclusion problem, which seeks a $z \in \mathbb{R}^d$ such that
$$0 \in Tz. \tag{4.1}$$
More compactly we will write the requirement as $z \in \operatorname{zer} T$. The preconditioned proximal point algorithm (PPPA) is given by
$$\begin{aligned} \bar{z}^k &= (P + T)^{-1} P z^k \\ z^{k+1} &= z^k + \lambda_k(\bar{z}^k - z^k) \end{aligned} \tag{PPPA}$$
where $T : \mathbb{R}^d \rightrightarrows \mathbb{R}^d$ is a set-valued operator and $P \in \mathbb{R}^{d \times d}$ is the preconditioning matrix.

The PPPA has been employed as an abstract framework to establish the convergence of numerical methods such as augmented Lagrangian method (ALM), progressive hedging, Douglas-Rachford splitting (DRS), various primal-dual methods, and many more (Rockafellar, 1976; Eckstein, 1988; Rockafellar & Sun, 2019; Eckstein & Bertsekas, 1992).

While convergence of DRS (upon which iFedDR and FedDR relies) is typically studied through nonexpansiveness properties of the resolvent (Bauschke & Combettes, 2017, Thm. 26.11), to allow for *relative* inexact analysis we shall cast it as an instance of PPPA with a *semidefinite* preconditioning. It is of immediate verification that DRS is nothing more than PPPA with the primal dual

operator $T = T_{\mathrm{PD}}$ as defined in (2.6) and the following preconditioning:

$$P = \begin{bmatrix} \gamma^{-1}I & -I \\ -I & \gamma I \end{bmatrix}. \tag{4.2}$$

Unlike the standard analysis of PPPA (Rockafellar, 1976) which assumes a positive definite preconditioning, here the preconditioner $P$ is only positive semidefinite. In recent years, PPPA with semidefinite preconditioner has been studied in (Latafat & Patrinos, 2017, Thm. 3.4), (Bredies et al., 2022, Sect. 2.1), (Evens et al., 2023, Thm. 2.4) and in the setting of DRS in (Condat, 2013, Thm. 3.3). However, existing results have been restricted to exact computations. Motivated by the above reformulation of DRS as PPPA, we develop an *inexactness* variant of PPPA with mild conditions on the inexactness suitable for federated learning applications. Our work is largely inspired by Solodov & Svaiter (1999) which proposed relative inexactness as an alternative to the standard *absolute summable* error criteria. In particular, we consider replacing (preconditioned) proximal updates by the following

$$\bar{z} = (P + T)^{-1}(Pz - \varepsilon) \iff Pz - P\bar{z} = \bar{v} + \varepsilon \text{ with } \bar{v} \in T\bar{z},$$

where error $\varepsilon \in \mathbb{R}^d$ would only be required to satisfy a relative condition waiving the summability assumption. Although Solodov & Svaiter (1999) studies the classical proximal point method with $P = \mathrm{I}$, their analysis carries over to the case where $P$ is positive definite. We here further generalize their analysis and study PPPA under mere positive *semidefiniteness* of the preconditioner. To this end we construct the positive definite matrix

$$R = \mathrm{I} - Q + P > 0, \quad \text{where} \quad Q = \Pi_{\mathrm{range}(P)},$$

where $\Pi_C(w) := \arg\min_{u \in C} \|u - w\|^2$ denotes the projection. While directly establishing convergence of the sequence $(z^k)_{k \in \mathbb{N}}$ through the machinery of Fejér monotonicity is no longer possible (due to lack of positive definiteness of $P$), we shall do so for the *shadow* sequence $(\Pi_{\mathrm{range}(P)}(z^k))_{k \in \mathbb{N}}$ in the space equipped with $\langle \cdot, \cdot \rangle_R$.

The proposed abstract inexact scheme corrects for the inaccuracy, $\varepsilon$, through an extragradient step that we will synonymously refer to as the *error-correction* step. The scheme is defined by the following iterates given $z^k \in \mathbb{R}^d$ and the error tolerance $\sigma \in (0, 1)$:

$$
\begin{aligned}
&\text{find} \quad \bar{z}^k \in \mathbb{R}^d \quad \text{and the associated} \quad \bar{v}^k \in T\bar{z}^k \cap \mathrm{range}\, P \\
&\text{s.t.} \quad Pz^k - P\bar{z}^k = \bar{v}^k + \varepsilon^k, \quad \|\varepsilon^k\|_{R^{-1}} \leq \sigma \max\{\|\bar{v}^k\|_{R^{-1}}, \|P\bar{z}^k - Pz^k\|_{R^{-1}}\} \\
&\text{compute} \quad z^{k+1} = z^k - \lambda_k \alpha_k P\bar{v}^k \quad \text{where} \quad \alpha_k = \frac{\langle \bar{v}^k, Pz^k - P\bar{z}^k \rangle_{R^{-1}}}{\|P\bar{v}^k\|_{R^{-1}}^2}
\end{aligned}
\tag{iPPPA}
$$

We will study iPPPA under the following assumptions.

**Assumption II** (Requirements for abstract iPPPA)**.**

 *(i) The operator $T : \mathbb{R}^d \rightrightarrows \mathbb{R}^d$ is maximally monotone, and the set of its zeros,* zer $T$*, is nonempty.*

 *(ii) The preconditioning matrix $P \in \mathbb{R}^{d \times d}$ is positive* semi*definite.*

We obtain the error-corrected algorithms, iFedDR and iDRS, as instances of iPPPA through the choice in (4.2). We remark that the requirement $\bar{v}^k \in \mathrm{range}\, P$ is a technical condition that is crucial in the positive semidefinite setting studied here. Nevertheless, this condition is satisfied by construction in iDRS and iFedDR and thus does not need to be verified. For detailed derivations see Appendix D. We emphasize that in the convergence analysis in the next section, $T$ and $P$ are not restricted to the particular choice made in (2.6) and (4.2) respectively, and are only required to satisfy Assumption II.

## 5 CONVERGENCE ANALYSIS

The convergence analysis of iPPPA hinges upon an indirect analysis by studying the properties of the shadow sequences $(w^k)_{k \in \mathbb{N}} = (Qz^k)_{k \in \mathbb{N}}$, and $(\bar{w}^k)_{k \in \mathbb{N}} = (Q\bar{z}^k)_{k \in \mathbb{N}}$ in place of the original sequences. Having computed the points $\bar{z}^k$ and $\bar{v}^k$ by solving the approximate resolvent in the first step of iPPPA, the update for evaluating $w^{k+1} = Qz^{k+1}$ can be viewed (refer to the proof of Theorem 5.1 for details)

as relaxed projections of $w^k$ onto the following halfspaces

$$\mathcal{D}_k = \left\{ u \mid \langle \bar{v}^k, P\bar{w}^k - Pu \rangle_{R^{-1}} \geq 0 \right\}. \tag{5.1}$$

The projection onto $\mathcal{D}_k$ is given by the simple update

$$\Pi_{\mathcal{D}_k}^{R^{-1}}(w) := \arg\min_{u \in \mathcal{D}_k} \|u - w\|_{R^{-1}}^2 = \begin{cases} w - \frac{\langle \bar{v}^k, Pw - P\bar{v}^k \rangle_{R^{-1}}}{\|P\bar{v}^k\|_{R^{-1}}^2} P\bar{v}^k & \text{if } w \notin \mathcal{D}_k \\ w & w \in \mathcal{D}_k. \end{cases} \tag{5.2}$$

It is essential to ensure that the set $Q \operatorname{zer} T$ is contained within $\mathcal{D}_k$. To see this note that since for any $w^\star \in Q \operatorname{zer} T$ there exists $z^\star \in \operatorname{zer} T$ such that

$$\langle \bar{v}^k, P\bar{w}^k - Pw^\star \rangle_{R^{-1}} = \langle \bar{v}^k, PQ\bar{w}^k - PQw^\star \rangle_{R^{-1}} \overset{(C.4)}{=} \langle \bar{v}^k, RQ\bar{w}^k - RQw^\star \rangle_{R^{-1}}$$

$$= \langle \bar{v}^k, Q\bar{z}^k - Qz^\star \rangle = \langle \bar{v}^k, \bar{z}^k - z^\star \rangle \geq 0,$$

where the last equality follows from $\bar{v}^k \in \operatorname{range} P$, and the inequality follows from monotonicity of $T$. Thus ensuring that $Q \operatorname{zer} T \subseteq \mathcal{D}_k$.

To ensure progress at every iteration (and thus to establish convergence), the crucial component in our analysis is to argue that $\langle \bar{v}^k, Pw^k - P\bar{w}^k \rangle_{R^{-1}}$ remains strictly positive (when $w^k \notin \mathcal{D}_k$) even with a semidefinite preconditioner $P$ as long as certain conditions holds for the inaccuracy $\varepsilon^k$. The precise requirement can be found in the update specified in iPPPA. The reader is referred to the proof of Theorem 5.1 for further details.

**Theorem 5.1** (Convergence and convergence rate analysis). *Suppose that Assumption II holds and that $\liminf_{k\to\infty}(2 - \lambda_k)\lambda_k > 0$. If for some $k \geq 0$, $\bar{z}^k \notin \operatorname{zer} T$ then $\alpha_k > 0$ is finite and $\|Pz^k - P\bar{z}^k\| \neq 0$. Moreover, either the algorithm terminates in finite number of iterations with $\bar{z}^{\bar{k}} \in \operatorname{zer} T$ for some $\bar{k} \geq 0$, or it generates an infinite sequence $(z^k)_{k\in\mathbb{N}}$ for which the following hold*

*(i) $(\|P\bar{z}^k - Pz^k\|)_{k\in\mathbb{N}}$ converges to zero.*

*(ii) $(\bar{z}^k)_{k\in\mathbb{N}}$ converges to some $z^\star \in \operatorname{zer} T$.*

*(iii) For all $z^\star \in \operatorname{zer} T$, the sequence $(\bar{z}^k)_{k\in\mathbb{N}}$ satisfies*

$$\min_{k=0,1,\ldots,m} \operatorname{dist}_{R^{-1}}^2(0, T\bar{z}^k) \leq \frac{\|Qz^0 - Qz^\star\|_{R^{-1}}^2}{\tau(m+1)}$$

*where $\tau = \liminf_{k\to\infty}(2 - \lambda_k)\lambda_k \frac{1}{\|R\|(1-\sigma)^2}$.*

**Remark 5.2.** Both iDRS and iFedDR are special cases of iPPPA so convergence of both schemes immediately follows from Theorem 5.1 (see Appendix D.1 and Appendix D.2 respectively). In particular, Theorem 3.1 follows as a corollary. $\qquad\square$

**Linear convergence** We additionally study local linear properties of iPPPA under *metric subregularity* (Dontchev & Rockafellar, 2009, Sect. 3) assumption of the operator $T$. This assumption amounts to requiring the distance from the set of solutions to be upper bounded by a multiple of the norm of $Tz$ for all points $z$ close to $z^\star$. This local property is a very mild assumption, and in particular does not imply uniqueness of the solution. Let us recall the notion of metric subregularity (Dontchev & Rockafellar, 2009, Sect. 3):

**Definition 5.3** (Metric subregularity). *A set-valued mapping $M : \mathbb{R}^n \rightrightarrows \mathbb{R}^d$ is* metrically subregular *at $\bar{x}$ for $\bar{y}$ if $(\bar{x}, \bar{y}) \in \operatorname{gph} M$ and there exists a positive constant $\eta$ together with a* neighborhood of subregularity $\mathcal{U}$ *of $\bar{x}$ such that*

$$\operatorname{dist}(x, M^{-1}\bar{y}) \leq \eta \operatorname{dist}(\bar{y}, Mx) \quad \forall x \in \mathcal{U}.$$

Metric subregularity has been used extensively in the optimization literature for establishing linear convergence for splitting techniques, see (Tseng, 2000; Drusvyatskiy & Lewis, 2018; Rockafellar, 2023).

**Theorem 5.4** (Linear convergence under metric subregularity). *Suppose that in addition to assumptions of Theorem 5.1, the operator $T$ is metrically subregular at all $z^\star \in \operatorname{zer} T$ for $0$. Then, the sequence $(\Pi_{\operatorname{range} P} z^k)_{k\in\mathbb{N}}$ generated by iPPPA converges R-linearly.*

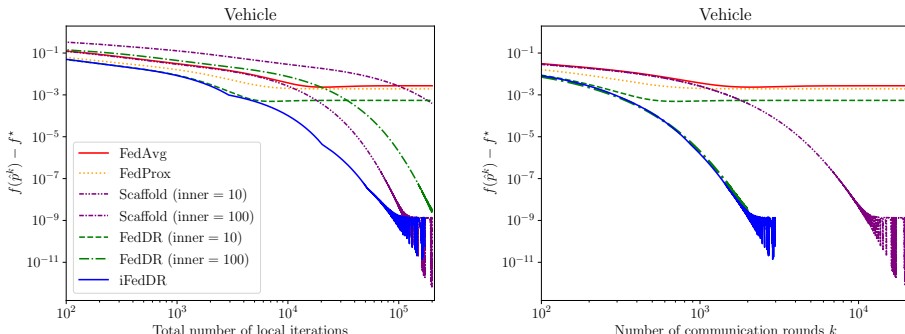

Figure 2: Logistic regression on a heterogeneous data split. We find that iFedDR can converge fast without requiring tuning due to the error-correction step and automatic refinement. In contrast, FedDR only converges when the number of inner iterations $\tau$ is set sufficiently high. Even without any tuning iFedDR is competitive with all the baselines for which both client stepsize and client steps are tuned.

## 6  EXPERIMENTS

In this section we demonstrate iFedDR on a range of numerical problems.

**Binary logistic regression**   We consider the vehicle dataset (Duarte & Hu, 2004) with the heterogenous split proposed for FL benchmarking in Wu et al. (2022) and a heterogenous split of the w8a dataset (Chang & Lin, 2011) as used in Mishchenko et al. (2022). The settings uses 23 clients and 10 clients respectively. We compare iFedDR against FedAvg, FedProx, FedDR and Scaffold (see Appendix E for exact definitions). The baselines are given an unfair advantage by gridsearching over the client stepsize and the number of inner iterations. In contrast, we run only a single configuration of iFedDR. See Table 2 in Appendix F for details on the hyperparameters. We compare both the total number of inner iterations and server iterations in Figure 2 (see Figure 5 for similar results on w8a). If we tune the stepsizes $\lambda$ and $\gamma$ in iFedDR, we can improve the convergence speed further as illustrated in Figure 8 of Appendix F and Figure 1.

**Linear probing**   A common practice is to take off-the-shelf pre-trained models and specialize them to a downstream task. Linear probing, in particular, freezes the feature mapping of a pre-trained model and trains a linear classifier on top. We follow the setup in (Nguyen et al., 2022), which uses a ResNet18 (He et al., 2016) pre-trained on ImageNet and for the downstream task considers a heterogeneous datasplit of CIFAR10 (Krizhevsky) across 20 clients with Dirichlet distribution with parameter 0.1. All methods except for iFedDR are tuned (cf. Table 3 in Appendix F for hyperparameters). The results can be found in Figure 3 where iFedDR either matches or surpasses the tuned baselines. We additionally train on Fashion-MNIST from scratch where we observe similar behavior (cf. Appendix F).

We observe that the number of refinements is monotonically increasing in the number of server steps. This observation motivates a very simple heuristic to almost entirely avoid the communication overhead caused by refinement. We simply increase the number of initially used inner iterations at the next server iteration $k$ to be $\tau_k = \tau \times \#(\text{total refinements})$. The rule still enjoys the convergence guarantees of Theorem 3.1, and, as long as the number of refinements are monotonically increasing, it does not increase the total number of client steps. As illustrated in Figure 3 on CIFAR10, iFedDR only needs 10 additional communication rounds *in total* using this simple rule. This can be further decreased if a doubling strategy is employed, which is especially useful in ill-conditioned problems.

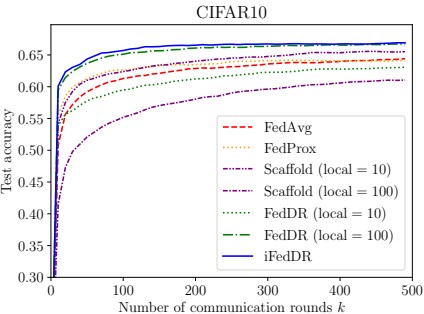 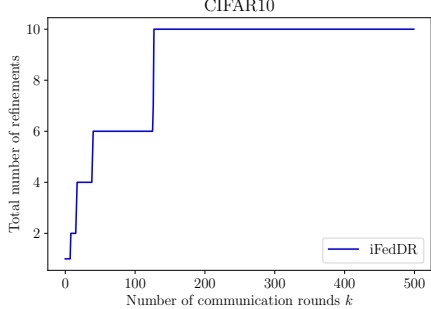

Figure 3: (left) Linear probing on CIFAR10 under heterogeneous data split. The iFedDR algorithm matches the test accuracy of a properly tuned FedDR algorithm. (right) By monotonically increasing the subsequent inner iteration when a refinement is triggered, iFedDR only uses 10 more communication rounds over the cause of the *entire* training run.

**Fair classification** To demonstrate the applicability to minimax problems, we consider the following formulation of fair classification (Sharma et al., 2022; Nouiehed et al., 2019):

$$\underset{u\in\mathbb{R}^m}{\text{minimize}} \ \underset{v\in\Delta_r}{\text{maximize}} \ \frac{1}{N}\sum_{i=1}^N v_i\ell_i(u) - \frac{\delta}{2}\|v\|^2 \qquad (6.1)$$

where $\Delta_r$ is the $r$-dimensional simplex and $\ell_i : \mathbb{R}^m \to \mathbb{R}$ is the cross entropy loss over the client dataset. We compare against baselines supporting projection. We use the same dataset split of CIFAR10 and model configuration as in Section 6 but report the test accuracy on the *worst class* in Figure 4 (see Table 4 for hyperparameters).

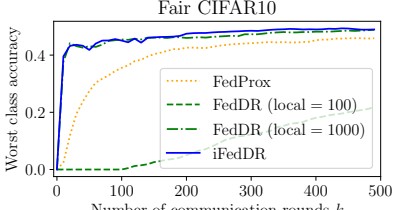

Figure 4: iFedDR on fair classification. iFedDR matches FedDR without sweeping over the number of inner steps.

## 7 CONCLUSION

We have developed a variant of FedDR, which automatically adjust the number of client iterations and consequently mitigate the need for hyperparameter selection. The scheme enjoys guarantees even for minimax problems (and more generally *m*-player games) and permits large proximal stepsizes. Convergence is proven for a much more general method, namely an error-corrected proximal point algorithm with a (general) semidefinite preconditioner, which might be interesting in its own right. We demonstrate the favorable properties of the algorithm on a range of numerical experiments. For future work it is interesting to extend the idea to partial participation and nonconvex problems.

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

# Appendix

## TABLE OF CONTENTS

## A    Related works

Federated learning (FL) has seen considerable attention since it was first coined in (McMahan et al., 2017). To treat heterogeneous data, (Li et al., 2020) proposed the prox-based method FedProx. Prox-based FL methods have since been developed by building on operator splitting techniques as in FedSplit (Pathak & Wainwright, 2020) and FedDR/FedPi (Tran Dinh et al., 2021; Malekmohammadi et al., 2022). See Pathak & Wainwright (2020) for a detailed discussion on the convergence-failure of pre-existing methods and Ryu & Yin (2022); Malekmohammadi et al. (2022) for an overview of splitting algorithms. The FL setting has since its conception been extended to fixed point problems (Malinovskiy et al., 2020) and minimax problem (Augenstein et al., 2019; Sharma et al., 2022; Ramezani-Kebrya et al., 2023).

The idea of using a hyperplane projection as a correction step was first introduced in Solodov & Svaiter (1999) for the proximal point method. It was later extended in Giselsson (2021) to treat a large family of operator splitting approaches in a unified fashion and have been used for proving convergence in nonmonotone problems (Pethick et al., 2023). A hyperplane projection argument was also used in the adaptive stepsize for finding the intersection of convex sets analyzed in Combettes (1997), which was recently applied to the FL setting in Jhunjhunwala et al. (2023).

## B    Preliminaries

We recall some standard definitions and refer to Bauschke & Combettes (2017); Rockafellar (1970) for further details. We will the denote the distance to a set $\mathcal{Z}$ under the positive definite matrix $D$ as $\text{dist}_D(z, \mathcal{Z}) := \min_{z' \in \mathcal{Z}} \|z - z'\|_D$ and the normal cone to $z \in \mathcal{Z}$ as $N_{\mathcal{Z}}(z) = \{v \in \mathbb{R}^n \mid \langle v, z' - z \rangle \leq 0 \; \forall z' \in \mathcal{Z}\}$. A sequence $(z^k)_{k \in \mathbb{N}}$ is said to be Fejér monotone with respect to a set $\mathcal{S} \subseteq \mathbb{R}^n$ if $\|z^{k+1} - z\| \leq \|z^k - z\|$ for all $z \in \mathcal{S}, k \in \mathbb{N}$.

An operator or set-valued mapping $A : \mathbb{R}^n \rightrightarrows \mathbb{R}^d$ maps each point $x \in \mathbb{R}^n$ to a subset $Ax$ of $\mathbb{R}^d$. We will use the notation $A(x)$ and $Ax$ interchangably. We denote the domain of $A$ by

$$\text{dom}\, A := \{x \in \mathbb{R}^n \mid Ax \neq \emptyset\},$$

its graph by

$$\text{gph}\, A := \{(x, y) \in \mathbb{R}^n \times \mathbb{R}^d \mid y \in Ax\},$$

and the set of its zeros by

$$\text{zer}\, A := \{x \in \mathbb{R}^n \mid 0 \in Ax\}.$$

The inverse of $A$ is defined through its graph: $\text{gph}\, A^{-1} := \{(y, x) \mid (x, y) \in \text{gph}\, A\}$. The *resolvent* of $A$ is defined by $J_A := (\text{id} + A)^{-1}$, where id denotes the identity operator. When $A$ is a subdifferential $\partial g$ of a closed, convex, and proper function $g : \mathbb{R}^n \to \mathbb{R}$, the resolvent reduces to the proximal operator, i.e. $(\text{id} + \gamma A)^{-1}(x) = \text{prox}_{\gamma g}(x) := \arg\min_{y \in \mathbb{R}^n} g(y) + \frac{1}{2\gamma}\|y - x\|^2$ for $\gamma > 0$.

**Definition B.1** (monotonicity). *An operator $A : \mathbb{R}^n \rightrightarrows \mathbb{R}^n$ is said to be monotone, if for all $(x, y), (x', y') \in \text{gph}\, A$*

$$\langle x - x', y - y' \rangle \geq 0.$$

*The operator $A$ is said to be maximally monotone if its graph is not strictly contained in the graph of another monotone operator.*

## C    The inexact preconditioned proximal point algorithm (iPPPA)

**Proof of Theorem 5.1** *(Convergence and convergence rate analysis).* Suppose that $Pz^k - P\bar{z}^k \neq 0$, since otherwise $0 = \bar{v}^k + \varepsilon^k$ which combined with $\|\varepsilon^k\|_{R^{-1}} \leq \sigma \max\{\|\bar{v}^k\|_{R^{-1}}, \|Pz^k - P\bar{z}^k\|_{R^{-1}}\} = \sigma\|\bar{v}^k\|_{R^{-1}}$ and $\sigma \in (0, 1)$, implies $\bar{v}^k = 0$, i.e., that $\bar{z}^k \in \text{zer}\, T$.

That $\alpha_k$ is finite follows from the fact that $\|\bar{v}^k\| = 0$ iff $\bar{z}^k \in \text{zer}\, T$.

We begin to derive a few basic relations that will be used in the sequel.

$$\|\bar{v}^k\|_{R^{-1}} = \|P(z^k - \bar{z}^k) - \varepsilon^k\|_{R^{-1}} \geq \|P(z^k - \bar{z}^k)\|_{R^{-1}} - \|\varepsilon^k\|_{R^{-1}} \geq \|P(z^k - \bar{z}^k)\|_{R^{-1}} - \sigma \max\left\{\|\bar{v}^k\|_{R^{-1}}, \|P\bar{z}^k - Pz^k\|_{R^{-1}}\right\}.$$
$$\tag{C.1}$$

Using the triangle inequality again

$$\|\bar{v}^k\|_{R^{-1}} \le \|Pz^k - P\bar{z}^k\|_{R^{-1}} + \|\varepsilon^k\|_{R^{-1}} \le \|Pz^k - P\bar{z}^k\|_{R^{-1}} + \sigma \max\left\{\|\bar{v}^k\|_{R^{-1}}, \|P\bar{z}^k - Pz^k\|_{R^{-1}}\right\} \quad \text{(C.2)}$$

It follows from the last two inequalities that

$$\tfrac{1}{\xi}\|\bar{v}^k\|_{R^{-1}} \le \|Pz^k - P\bar{z}^k\|_{R^{-1}} \le \xi\|\bar{v}^k\|_{R^{-1}}, \quad \text{(C.3)}$$

where $\xi = \max\left\{1 + \sigma, \frac{1}{1-\sigma}\right\} = \frac{1}{1-\sigma}$ since $\sigma < 0$.

From the construction of $R$ it follows that

$$P = R \circ Q = P \circ Q = Q \circ P. \quad \text{(C.4)}$$

We have the following identity

$$\|P\bar{v}^k\|_{R^{-1}}^2 = \langle P\bar{v}^k, P\bar{v}^k \rangle_{R^{-1}} = \langle PQ\bar{v}^k, PQ\bar{v}^k \rangle_{R^{-1}} = \langle \bar{v}^k, R\bar{v}^k \rangle = \|\bar{v}^k\|_R^2, \quad \text{(C.5)}$$

where $\bar{v}^k \in \text{range } P$ was used in the third equality.

We start by introducing a shadow sequences $w^k = Qz^k$, $\bar{w}^k = Q\bar{z}^k$ and defining the halfspace

$$\mathcal{D}_k = \left\{u \mid \langle \bar{v}^k, P\bar{w}^k - Pu \rangle_{R^{-1}} \ge 0\right\}. \quad \text{(C.6)}$$

We have that $Q \text{ zer } T \subseteq \mathcal{D}_k$ since for any $w^\star \in Q \text{ zer } T$ there exists $z^\star \in \text{zer } T$ such that

$$\langle \bar{v}^k, P\bar{w}^k - Pw^\star \rangle_{R^{-1}} = \langle \bar{v}^k, PQ\bar{w}^k - PQw^\star \rangle_{R^{-1}}$$
$$\text{(C.4)} = \langle \bar{v}^k, RQ\bar{w}^k - RQw^\star \rangle_{R^{-1}}$$
$$= \langle \bar{v}^k, Q\bar{z}^k - Qz^\star \rangle$$
$$= \langle \bar{v}^k, \bar{z}^k - z^\star \rangle \ge 0,$$

where the last equality follows from $\bar{v}^k \in \text{range } P$, and the inequality follows from monotonicity of $T$.

Suppose that $w^k \notin \mathcal{D}_k$ for the moment. The alternative will be dealt with later. Then, it follows from Fact C.1 that

$$\Pi_{\mathcal{D}_k}^{R^{-1}}(w^k) = w^k - \frac{\langle \bar{v}^k, Pw^k - P\bar{w}^k \rangle_{R^{-1}}}{\|P\bar{v}^k\|_{R^{-1}}^2} P\bar{v}^k.$$

Note that

$$w^{k+1} = Qz^{k+1} = Qz^k - \lambda_k\alpha_k QP\bar{v}^k = w^k - \lambda_k\alpha_k P\bar{v}^k = (1 - \lambda_k)w^k + \lambda_k \Pi_{\mathcal{D}_k}^{R^{-1}}(w^k).$$

Recall that the mapping $(1 - \lambda_k)\text{id} + \lambda_k \Pi_{\mathcal{D}_k}^{R^{-1}}$ is $\lambda_k/2$-averaged in the space equiped with $\langle \cdot, \cdot \rangle_{R^{-1}}$. Hence, the sequence $(w^k)_{k \in \mathbb{N}}$ is Fejér monotone relative to $Q \text{ zer } T$, i.e.,

$$\|w^{k+1} - w^\star\|_{R^{-1}}^2 \le \|w^k - w^\star\|_{R^{-1}}^2 - \frac{1 - \frac{\lambda_k}{2}}{\frac{\lambda_k}{2}}\|w^{k+1} - w^k\|^2$$
$$= \|w^k - w^\star\|_{R^{-1}}^2 - (2 - \lambda_k)\lambda_k\alpha_k^2\|P\bar{v}^k\|_{R^{-1}}^2 \quad \text{(C.7)}$$

We proceed to consider two cases.

♠ Case 1: $\max\{\|\bar{v}^k\|_{R^{-1}}, \|Pz^k - P\bar{z}^k\|_{R^{-1}}\} = \|\bar{v}^k\|_{R^{-1}}$.

We have

$$\langle \bar{v}^k, Pz^k - P\bar{z}^k \rangle_{R^{-1}} = \langle \bar{v}^k, \bar{v}^k + \varepsilon^k \rangle_{R^{-1}}$$
$$\ge \|\bar{v}^k\|_{R^{-1}}^2 - \|\bar{v}^k\|_{R^{-1}}\|\varepsilon^k\|_{R^{-1}} = \|\bar{v}^k\|_{R^{-1}}\left(\|\bar{v}^k\|_{R^{-1}} - \|\varepsilon^k\|_{R^{-1}}\right)$$
$$\ge (1 - \sigma)\|\bar{v}^k\|_{R^{-1}}^2 \quad \text{(C.8)}$$
$$\ge \frac{(1-\sigma)}{\xi}\|P(z^k - \bar{z}^k)\|_{R^{-1}}\|\bar{v}^k\|_{R^{-1}} \quad \text{(C.9)}$$

where (C.3) was used in the last inequality.

♠ Case 2: $\max\{\|\bar{v}^k\|_{R^{-1}}, \|Pz^k - P\bar{z}^k\|_{R^{-1}}\} = \|Pz^k - P\bar{z}^k\|_{R^{-1}}$.

Similarly, we have

$$\langle \bar{v}^k, Pz^k - P\bar{z}^k \rangle_{R^{-1}} = \|Pz^k - P\bar{z}^k\|_{R^{-1}}^2 - \langle \varepsilon^k, Pz^k - P\bar{z}^k \rangle_{R^{-1}}$$
$$\ge \|Pz^k - P\bar{z}^k\|_{R^{-1}}^2 - \|Pz^k - P\bar{z}^k\|_{R^{-1}}\|\varepsilon^k\|_{R^{-1}}$$

$$\geq \tfrac{(1-\sigma)}{\xi}\|P(z^k - \bar{z}^k)\|_{R^{-1}}\|\bar{v}^k\|_{R^{-1}}, \tag{C.10}$$

where (C.3) was used in the last inequality.

In either case, we have

$$\langle \bar{v}^k, Pz^k - P\bar{z}^k \rangle_{R^{-1}} \geq \tfrac{(1-\sigma)}{\xi}\|P(z^k - \bar{z}^k)\|_{R^{-1}}\|\bar{v}^k\|_{R^{-1}}. \tag{C.11}$$

If $w^k \in \mathcal{D}_k$, then

$$0 \geq \langle \bar{v}^k, Pw^k - P\bar{w}^k \rangle_{R^{-1}} = \langle \bar{v}^k, Pz^k - P\bar{z}^k \rangle_{R^{-1}} \geq \tfrac{(1-\sigma)}{\xi}\|P(z^k - \bar{z}^k)\|_{R^{-1}}\|\bar{v}^k\|_{R^{-1}}$$

which implies $\bar{z}^k \in \operatorname{zer} T$ since it must be that $\|\bar{v}^k\|_{R^{-1}} = 0$.

Suppose that $w^k \notin \mathcal{D}_k$. Then,

$$\alpha_k\|P\bar{v}^k\|_{R^{-1}} = \frac{\langle \bar{v}^k, Pz^k - P\bar{z}^k \rangle_{R^{-1}}}{\|P\bar{v}^k\|_{R^{-1}}^2}\|P\bar{v}^k\|_{R^{-1}}$$

$$\text{(C.11)} \geq \tfrac{(1-\sigma)}{\xi}\|P(z^k - \bar{z}^k)\|_{R^{-1}}\frac{\|\bar{v}^k\|_{R^{-1}}}{\|P\bar{v}^k\|_{R^{-1}}}$$

$$\text{(C.5)} = \tfrac{(1-\sigma)}{\xi}\|P(z^k - \bar{z}^k)\|_{R^{-1}}\frac{\|\bar{v}^k\|_{R^{-1}}}{\|\bar{v}^k\|_R} \geq c_2\tfrac{(1-\sigma)}{\xi}\|P(z^k - \bar{z}^k)\|_{R^{-1}} \tag{C.12}$$

for $c_2 = 1/\sqrt{\|R\|} > 0$, where we have used that the Rayleigh quotient, $\frac{\|\bar{v}^k\|_{R^{-1}}^2}{\|\bar{v}^k\|^2}$, can be lower bounded by $\lambda_{\min}(R^{-1}) = 1/\|R\|$ where $\lambda_{\min}(R^{-1})$ denotes the minimum eigenvalue of $R^{-1}$.

We showed that if $w^k \in \mathcal{D}_k$, the algorithm has terminated. Otherwise, if $w^k \notin \mathcal{D}_k$, using (C.12) in (C.7) yields

$$\|w^{k+1} - w^\star\|_{R^{-1}}^2 \leq \|w^k - w^\star\|_{R^{-1}}^2 - (2 - \lambda_k)\lambda_k\tfrac{c_2^2(1-\sigma)^2}{\xi^2}\|P\bar{z}^k - Pz^k\|_{R^{-1}}^2 \tag{C.13}$$

$$\leq \|w^k - w^\star\|_{R^{-1}}^2 - (2 - \lambda_k)\lambda_k\tfrac{c_2^2(1-\sigma)^2}{\xi^4}\|\bar{v}^k\|_{R^{-1}}^2. \tag{C.14}$$

It follows that $(w^k)_{k\in\mathbb{N}}$ is bounded. In turn, implying boundedness of $(\bar{z}^k)_{k\in\mathbb{N}}$ after observing that

$$\bar{z}^k = (P + T)^{-1}Pz^k = (P + T)^{-1}Pw^k, \tag{C.15}$$

and that the preconditioned resolvent is locally bounded. Moreover, using a telescoping argument we have that $(\|\bar{v}^k\|)_{k\in\mathbb{N}}$ converges to zero, implying that the limit points of $\bar{z}^k$ belong to $\operatorname{zer} T$ as claimed. Hence, the limit points of $(\bar{w}^k)_{k\in\mathbb{N}} = (Q\bar{z}^k)_{k\in\mathbb{N}}$ belong to $Q\operatorname{zer} T$. As another consequence of (C.13), we have

$$\textstyle\sum_{k=0}^\infty \|\bar{w}^k - w^k\|_R^2 = \sum_{k=0}^\infty \|P\bar{z}^k - Pz^k\|_{R^{-1}}^2 < \infty, \tag{C.16}$$

which implies that the limit points of $(w^k)_{k\in\mathbb{N}}$ belong to $Q\operatorname{zer} T$ as well. The convergence of $(w^k)_{k\in\mathbb{N}}$ then follows by (Bauschke & Combettes, 2017, Thm. 5.5). In turn by continuity of the preconditioned resolvent and (C.15) the convergence of $(\bar{z}^k)_{k\in\mathbb{N}}$ follows.

The rate follows by telescoping (C.14) to obtain

$$\tfrac{1}{m+1}\textstyle\sum_{k=0}^m \|\bar{v}^k\|_{R^{-1}}^2 \leq \frac{\|w^0 - w^\star\|_{R^{-1}}^2}{\tau(m+1)} \tag{C.17}$$

with $\tau = \liminf_{k\to\infty}(2 - \lambda_k)\lambda_k c_2^2(1-\sigma)^2/\xi^4$. Observing that $\operatorname{dist}_{R^{-1}}(0, T\bar{z}^k) := \min_{\bar{u}^k \in T\bar{z}^k}\|\bar{u}^k\|_{R^{-1}} \leq \|\bar{v}^k\|_{R^{-1}}$ completes the proof. □

**Fact C.1.** *Given a point $r \notin \mathcal{D}_k$, the projection onto $\mathcal{D}_k$ is given by*

$$\Pi_{\mathcal{D}_k}^{R^{-1}}(r) = r - \frac{\langle \bar{v}^k, Pr - P\bar{w}^k \rangle_{R^{-1}}}{\|P\bar{v}^k\|_{R^{-1}}^2}P\bar{v}^k.$$

**_Proof of Theorem 5.4_** *(Linear convergence under metric subregularity).* The proof relies on establishing a contraction by exploiting metric subregularity assumption and builds upon the analysis of (Latafat & Patrinos, 2017, Thm. 3.3) extending it to the inexact preconditioned proximal point setting with positive semidefinite preconditioning.

Recal that $Q = \operatorname{range} P$ and note that

$$P = R \circ Q = P \circ Q = Q \circ P. \tag{C.18}$$

Recall that by the descent inequality for the shadow sequence (C.13):

$$\text{dist}_{R^{-1}}^2(w^{k+1}, Q \text{ zer } T) \leq \|w^{k+1} - \Pi_{Q \text{ zer } T}^{R^{-1}}(w^k)\|_{R^{-1}}^2$$

$$\leq \|w^k - \Pi_{Q \text{ zer } T}^{R^{-1}}(w^k)\|_{R^{-1}}^2 - (2 - \lambda_k)\lambda_k \frac{c_2^2(1-\sigma)^2}{\xi^2}\|P\bar{z}^k - Pz^k\|_{R^{-1}}^2$$

$$= \text{dist}_{R^{-1}}^2(w^k, Q \text{ zer } T) - (2 - \lambda_k)\lambda_k \frac{c_2^2(1-\sigma)^2}{\xi^2}\|P\bar{z}^k - Pz^k\|_{R^{-1}}^2 \qquad \text{(C.19)}$$

where in the last equality (C.18) was used, while attainment of the projection is obtained by convexity of zer $T$.

We proceed to establish a contraction in terms of the above distance sequence by lower bounding the last term. By Theorem 5.1 the sequence $\bar{z}^k$ converges to some $z^\star \in$ zer $T$. Therefore, up to discarding initial iterates, the whole sequence lies within the neighborhood of metric subregularity. Since $\bar{v}^k \in T\bar{z}^k$, it follows from the subregularity assumption that there exists some $\eta > 0$ such that

$$\text{dist}(\bar{z}^k, \text{zer } T) \leq \eta\|\bar{v}^k\|. \qquad \text{(C.20)}$$

On the other hand, for the shadow sequence

$$\text{dist}_{R^{-1}}(\bar{w}^k, Q \text{ zer } T) = \inf_{v \in \text{zer } T} \|Qv - \bar{w}^k\|_{R^{-1}} = \inf_{v \in \text{zer } T} \|Qv - Q\bar{z}^k\|_{R^{-1}}$$

$$\leq \inf_{v \in \text{zer } T} \|v - \bar{z}^k\|_{R^{-1}} = \text{dist}_{R^{-1}}(\bar{z}^k, \text{zer } T)$$

$$\leq \|R^{-1}\|^{1/2} \text{dist}(\bar{z}^k, \text{zer } T)$$

$$\leq \eta\|R^{-1}\|^{1/2}\|\bar{v}^k\|$$

$$\leq \eta\|R^{-1}\|^{1/2}\|R\|^{1/2}\|\bar{v}^k\|_{R^{-1}} \qquad \text{(C.21)}$$

where (C.20) was used in the second last inequality while $\|Q\| = 1$ was used in the first one.

$$\text{dist}_{R^{-1}}(w^k, Q \text{ zer } T) \leq \|w^k - \Pi_{Q \text{ zer } T}(\bar{w}^k)\|_{R^{-1}} \leq \|\bar{w}^k - \Pi_{Q \text{ zer } T}(\bar{w}^k)\|_{R^{-1}} + \|\bar{w}^k - w^k\|_{R^{-1}}$$

$$= \text{dist}_{R^{-1}}(\bar{w}^k, Q \text{ zer } T) + \|\bar{w}^k - w^k\|_{R^{-1}}$$

$$\overset{\text{(C.21)}}{\leq} \eta\|R^{-1}\|^{1/2}\|R\|^{1/2}\|\bar{v}^k\|_{R^{-1}} + \|\bar{w}^k - w^k\|_{R^{-1}}$$

$$\overset{\text{(C.18)}}{\leq} \eta\|R^{-1}\|^{1/2}\|R\|^{1/2}\|\bar{v}^k\|_{R^{-1}} + \|R^{-1}\| \|P\bar{z}^k - Pz^k\|_{R^{-1}}$$

$$\overset{\text{(C.3)}}{\leq} \|R^{-1}\|^{1/2}(\eta\xi\|R\|^{1/2} + \|R^{-1}\|^{1/2})\|P\bar{z}^k - Pz^k\|_{R^{-1}}.$$

Combined with (C.19), $Q$-linear convergence of $\text{dist}_{R^{-1}}(w^k, Q \text{ zer } T)$ follows. In turn, by rearranging (C.19), $R$-linear convergence of $(\|Pz^k - P\bar{z}^k\|_{R^{-1}}^2)_{k \in \mathbb{N}}$ and consequently that of $(\|\bar{v}^k\|_{R^{-1}}^2)_{k \in \mathbb{N}}$ follows by (C.3). In light of finite $\alpha_k > 0$ (due to Theorem 5.1) and since $\|w^{k+1} - w^k\|_{R^{-1}} = \lambda_k\alpha_k\|P\bar{v}^k\|_{R^{-1}}$ and (C.12), the claimed rate for $(\|w^{k+1} - w^k\|_{R^{-1}}^2)_{k \in \mathbb{N}}$ and the shadow sequence $(w^k)_{k \in \mathbb{N}}$ follows. □

# D  SPECIAL CASES OF IPPPA

## D.1  INEXACT DOUGLAS-RACHFORD SPLITTING

Consider the following inclusion problem which seeks $x \in \mathbb{R}^{d/2}$ such that

$$0 \in (A + B)x \qquad \text{(D.1)}$$

where $A : \mathbb{R}^{d/2} \rightrightarrows \mathbb{R}^{d/2}$ and $B : \mathbb{R}^{d/2} \rightrightarrows \mathbb{R}^{d/2}$ are maximally monotone. As noted in Section 4, DRS applied to problem (D.1) can be written as an instance of PPPA.

We instead consider an inexact version of PPPA with correction, which we recall for convenience while dropping the iteration counter $k$. Given $z \in \mathbb{R}^d$, one iteration of the scheme proceeds as follows:

$$\begin{array}{ll}
\text{find} & \bar{z} \in \mathbb{R}^d \quad \text{and the associated} \quad \bar{v} \in T\bar{z} \cap \text{range } P \\
\text{s.t.} & Pz - P\bar{z} = \bar{v} + \varepsilon, \quad \|\varepsilon\|_{R^{-1}} \leq \sigma \max\{\|\bar{v}\|_{R^{-1}}, \|P\bar{z} - Pz\|_{R^{-1}}\} \\
\text{compute} & z^+ = z - \lambda\alpha P\bar{v} \quad \text{where } \alpha = \dfrac{\langle \bar{v}, Pz - P\bar{z}\rangle_{R^{-1}}}{\|P\bar{v}\|_{R^{-1}}^2}
\end{array} \qquad \text{(iPPPA)}$$

To derive the iDRS algorithm we apply iPPPA to the primal dual problem (2.6) by taking the operator $T = T_{\text{PD}}$ and choose the preconditioner $P$ as defined in (4.2). Note that the following derivation works for any maximally monotone operators $A, B$, and the operators do not necessarily have to be chosen as in (2.4b). We will later in Appendix D.2 specialize to (2.4b) involving the consensus constraint to derive the iFedDR algorithm. The iPPPA reduces to

$$\bar{z} = (P + T)^{-1}(Pz - \varepsilon) \quad \Leftrightarrow \quad Pz - P\bar{z} \in T\bar{z} + \varepsilon$$

$$\Leftrightarrow \quad \begin{bmatrix} \gamma^{-1}x - y \\ \gamma y - x \end{bmatrix} - \begin{bmatrix} \gamma^{-1}\bar{x} - \bar{y} \\ \gamma \bar{y} - \bar{x} \end{bmatrix} \in \begin{bmatrix} A\bar{x} + \bar{y} \\ B^{-1}\bar{y} - \bar{x} \end{bmatrix} + \begin{bmatrix} \varepsilon_1 \\ \varepsilon_2 \end{bmatrix}$$

with $\varepsilon = (\varepsilon_1, \varepsilon_2)$ and $\bar{z} = (\bar{x}, \bar{y})$. By picking $s = x - \gamma y$ the update for $\bar{x}$ reduces to

$$\bar{x} = (\text{id} + \gamma A)^{-1}(s - \gamma \varepsilon_1) \quad \Leftrightarrow \quad \bar{x} = s - \gamma(\bar{u} + \varepsilon_1) \quad \text{with} \quad \bar{u} \in A\bar{x}. \tag{D.2}$$

With $\bar{s} := \bar{x} - \gamma \bar{y}$ and $\bar{v} = (\bar{v}_1, \bar{v}_2)$, we further have

$$T\bar{z} = \begin{bmatrix} A\bar{x} + \bar{y} \\ B^{-1}\bar{y} - \bar{x} \end{bmatrix} \ni \begin{bmatrix} \bar{v}_1 \\ \bar{v}_2 \end{bmatrix} = \begin{bmatrix} \gamma^{-1}(s - \bar{s}) - \varepsilon_1 \\ \bar{s} - s - \varepsilon_2 \end{bmatrix} \quad \text{and} \quad Pz - P\bar{z} = \begin{bmatrix} \gamma^{-1}(s - \bar{s}) \\ \bar{s} - s \end{bmatrix}. \tag{D.3}$$

The range of $P$ consists of vectors of the form $(a, -\gamma a)$ for some $a \in \mathbb{R}^{d/2}$. The requirement $\bar{v} \in \text{range } P$ implies that $\bar{v}$ must have the structure $\bar{v}_2 = -\gamma \bar{v}_1$. Thus, it follows from the inclusion in (D.3) that the update for $\bar{y}$ reduces to

$$\bar{y} = (\text{id} + B^{-1})^{-1}(\gamma^{-1}s) = \gamma^{-1}s - \gamma^{-1}(\text{id} + \gamma B)^{-1}s \tag{D.4}$$

where the last equality uses the resolvent identity (Rockafellar & Wets, 2011, Lm. 12.14),

$$\beta(\text{id} + \beta^{-1}M)^{-1}x + (\text{id} + \beta M^{-1})^{-1}(\beta x) = \beta x. \tag{D.5}$$

which holds when the operator $M : \mathbb{R}^{d/2} \rightrightarrows \mathbb{R}^{d/2}$ is maximally monotone and $\beta > 0$.

The requirement $\bar{v} \in \text{range } P$ puts a restriction on $\varepsilon$, which can be characterized by developing the inclusion in (D.3)

$$\begin{bmatrix} -\gamma A\bar{x} - \gamma \bar{y} \\ B^{-1}\bar{y} - \bar{x} \end{bmatrix} \ni \begin{bmatrix} \bar{s} - s + \gamma \varepsilon_1 \\ \bar{s} - s - \varepsilon_2 \end{bmatrix} \quad \Leftrightarrow \quad \gamma^{-1}(s - (\gamma \varepsilon_1 + \varepsilon_2)) \in \bar{y} + \gamma^{-1}B^{-1}\bar{y}$$

$$\Leftrightarrow \quad \bar{y} = (\text{id} + B^{-1})^{-1}(\gamma^{-1}(s - (\gamma \varepsilon_1 + \varepsilon_2)))$$

This together with (D.4) in turn implies that $\varepsilon_2 = -\gamma \varepsilon_1$.

Note that

$$\bar{v}_2 = -\gamma \bar{v}_1 = -\gamma(\bar{u} + \bar{y}).$$

We are now ready to compute the following quantities appearing in iPPPA

$$\|Pz - P\bar{z}\|_{R^{-1}} = \frac{1}{\sqrt{\gamma}}\|\bar{s} - s\|$$

$$\|\varepsilon\|_{R^{-1}} = \sqrt{\gamma}\|\varepsilon_1\|$$

$$\|\bar{v}\|_{R^{-1}} = \sqrt{\gamma}\|\bar{u} + \bar{y}\|$$

$$\|P\bar{v}\|_{R^{-1}} = \frac{1+\gamma^2}{\sqrt{\gamma}}\|\bar{u} + \bar{y}\|$$

$$\langle \bar{v}, Pz - P\bar{z} \rangle_{R^{-1}} = \langle \bar{u} + \bar{y}, s - \bar{s} \rangle$$

All that remains is to write down the update rule for the next iterate $s^+$,

$$s^+ = x^+ - \gamma y^+ = x - \lambda \alpha \bar{v}_1 - \gamma y + \gamma \lambda \alpha \bar{v}_2 = s - \lambda \alpha \frac{(1+\gamma^2)^2}{\gamma}(\bar{u} + \bar{y}),$$

where we have used that

$$\begin{bmatrix} x^+ \\ y^+ \end{bmatrix} = \begin{bmatrix} x \\ y \end{bmatrix} - \lambda \alpha P\bar{v} = \begin{bmatrix} x - \lambda \alpha(\frac{1}{\gamma} + \gamma)(\bar{u} + \bar{y}) \\ y + \lambda \alpha(1 + \gamma^2)(\bar{u} + \bar{y}) \end{bmatrix}.$$

We can now define the inexact DRS algorithm (iDRS) specified in Algorithm II by specializing iPPPA. We absorb the $\gamma$-dependent factor into the adaptive stepsize parameter by setting $\alpha_k = \alpha \frac{(1+\gamma^2)^2}{\gamma}$, since the constants turns out to cancel out.

To provide convergence guarantees for the iDRS algorithm, we will generalize the *natural residual* defined in (3.2) to allow for the operator $A$ to be possibly set-valued:

$$\mathcal{G}_\gamma(\bar{x}^k) := \frac{1}{\gamma}(\bar{x}^k - (\text{id} + \gamma B)^{-1}(\bar{x}^k - \gamma \bar{u}^k)) \tag{D.8}$$

---

**Algorithm II** Inexact DRS (iDRS)

---

REQUIRE   $s^0 \in \mathbb{R}^{d/2}$ $\lambda \in (0, 2)$, $\gamma \in (0, \infty)$, $\sigma \in (0, 1)$

REPEAT for $k = 0, 1, \ldots$ until convergence

  **II**.1:  Find

$$\bar{x}^k = s^k - \gamma(\bar{u}^k + \varepsilon_1^k) \quad \text{and the associated} \quad \bar{u}^k \in A\bar{x}^k \tag{D.6}$$

$$\bar{y}^k = \gamma^{-1}(\bar{x}^k - \gamma\bar{u}^k) - \gamma^{-1}(\text{id} + \gamma B)^{-1}(\bar{x}^k - \gamma\bar{u}^k) \tag{D.7}$$

$$\bar{s}^k = \bar{x}^k - \gamma\bar{y}^k$$

such that

$$\|\varepsilon_1^k\| \le \sigma \max\left\{\|\bar{u}^k + \bar{y}^k\|, \tfrac{1}{\gamma}\|\bar{s}^k - s^k\|\right\}$$

  **II**.2:

$$s^{k+1} = s^k - \lambda\alpha_k(\bar{u}^k + \bar{y}^k) \quad \text{with} \quad \alpha_k = \frac{\langle \bar{u}^k + \bar{y}^k, s^k - \bar{s}^k \rangle}{\|\bar{u}^k + \bar{y}^k\|^2}$$

RETURN  $s^{k+1}$

---

where $\bar{u}^k$ is the point in the set $A\bar{x}^k$ as defined in (D.6). It is of immediate verification that the natural residual $\mathcal{G}_\gamma(\bar{x}^k)$ vanishes if and only if $\bar{x} \in \text{zer}(A + B)$.

The iDRS algorithm enjoys the following convergence guarantee.

**Theorem D.1.** *Suppose that $A : \mathbb{R}^{d/2} \rightrightarrows \mathbb{R}^{d/2}$ and $B : \mathbb{R}^{d/2} \rightrightarrows \mathbb{R}^{d/2}$ are maximally monotone and that $\text{zer}(A + B) \ne \emptyset$. Let $T_{\text{PD}} : \mathbb{R}^d \rightrightarrows \mathbb{R}^d$ as in (2.6) and let $(\bar{x}^k)_{k \in \mathbb{N}}$ and $(\bar{y}^k)_{k \in \mathbb{N}}$ be generated by iDRS (Algorithm II). Then,*

  *(i) The iterates $(\bar{x}^k, \bar{y}^k)_{k \in \mathbb{N}}$ converges to some $(x^\star, y^\star) \in \text{zer } T_{\text{PD}}$.*

  *(ii) For all $s^\star = x^\star - \gamma y^\star$ where $(x^\star, y^\star) \in \text{zer } T_{\text{PD}}$, we have that*

$$\min_{k=0,1,\ldots,m} \|\mathcal{G}_\gamma(\bar{x}^k)\|^2 \le \frac{\|s^0 - s^\star\|^2}{\tau(m + 1)} \tag{D.9}$$

  *where $\tau = \frac{(2-\lambda)\lambda\gamma(1+\gamma^2)}{(1-\sigma)^2}$.*

*Proof.* The claim is obtain as a special case of Theorem 5.1*(iii)* where maximal monotonicity of $T = T_{\text{PD}}$ (Assumption II*(i)*) in Theorem 5.1 is implied by that of the operators $A$ and $B$. Assumption II*(ii)* is satisfied by the choice of the preconditioner $P$ in (4.2). Theorem D.1*(i)* follows directly from Theorem 5.1*(ii)* with $T = T_{\text{PD}}$.

To establish the convergence rate in terms of variables of iDRS we will specialize the $\|\bar{v}^k\|^2_{R^{-1}}$ appearing in the rate (C.17):

$$\|\bar{v}^k\|^2 = \|\bar{v}_1^k\|^2 + \|\bar{v}_2^k\|^2 = (1 + \gamma^2)\|\mathcal{G}_\gamma(\bar{x}^k)\|^2 \tag{D.10}$$

where we have used that

$$\bar{v}_2^k = -\gamma\bar{v}_1^k = -\gamma(\bar{u}^k + \bar{y}^k) = \gamma\mathcal{G}_\gamma(\bar{x}^k)$$

where the last equality follows from (D.7) by recognizing the definition of the natural residual (D.8). Furthermore, we have

$$\|\bar{v}^k\|^2_{R^{-1}} \ge \tfrac{1}{\|R\|}\|\bar{v}^k\|^2 = \tfrac{\gamma}{1+\gamma^2}\|\bar{v}^k\|^2 \tag{D.11}$$

where the last equality follows from the particular choice of the preconditioner $P$ in (4.2).

Combining (D.10) and (D.11) we get

$$\gamma\|\mathcal{G}_\gamma(\bar{x}^k)\|^2 \le \tfrac{\gamma}{1+\gamma^2}\|\bar{v}^k\|^2 \le \|\bar{v}^k\|^2_{R^{-1}} \tag{D.12}$$

What remains is to specialize the initial distance $\|Qz^0 - Qz^\star\|^2_{R^{-1}}$. From the particular choice of the preconditioner $P$, we have that

$$Q(z) := \Pi_{\text{range}(P)}(z) = \frac{1}{1+\gamma^2}\begin{bmatrix} s \\ -\gamma s \end{bmatrix}$$

with $z = (x, y)$ and $s = x - \gamma y$. It follows that

$$\|Qz^0 - Qz^\star\|_{R^{-1}}^2 \le \|R^{-1}\| \, \|Qz^0 - Qz^\star\|^2 = \frac{\|R^{-1}\|}{1+\gamma^2}\|s^0 - s^\star\|^2 \qquad (D.13)$$

where $s^\star = x^\star - \gamma y^\star$ with $(x^\star, y^\star) \in \operatorname{zer} T_{\mathrm{PD}}$. By substituting the computed quantities (D.12) and (D.13) into the rate (C.17) and choosing constant $\lambda_k = \lambda$ for simplicity, we obtain

$$\min_{k=0,1,\dots,m} \gamma\|\mathcal{G}_\gamma(\bar{x}^k)\|^2 \le \frac{\|R\| \, \|R\|^{-1}(1 - \sigma)^2\|s^0 - s^\star\|^2}{(2 - \lambda)\lambda(1 + \gamma^2)(m + 1)}$$

Rearranging and noting that $\|R\| \, \|R^{-1}\| = 1$ completes the proof. $\qquad\square$

### D.2 The inexact federated Douglas-Rachford algorithm

The iDRS algorithm developed in Appendix D.1 can further be specified to the federated learning setting by making the particular choice of the operators $A, B$ provided in (2.4b). Note that the resulting inclusion problem (2.4) is an equivalent reformulation of the original federated learning problem (2.1) as covered by Section 2. The reformulation introduces a lifted space whose variable will be indicated by bold font, e.g. $\boldsymbol{x} \in \mathbb{R}^{Nn}$.

The iDRS algorithm requires computing the resolvent $(\mathrm{id} - \gamma B)^{-1}$ which acts in the lifted space. The computation simplifies under the particular choice of the operator $B$ in (2.4b) involving the consensus constraint by reducing to computing the resolvent of $G$ on an average as made precise in Lemma D.2. See e.g. Latafat et al. (2021, Lm. 3.1) for a similar result in the specific case of minimization.

**Lemma D.2.** *Let $B : \mathbb{R}^{Nn} \rightrightarrows \mathbb{R}^{Nn}$ be defined as in (2.4b) and $\gamma > 0$. Then, given $\boldsymbol{x} = (x_1, \dots, x_N) \in \mathbb{R}^{Nn}$, the resolvent of $B$ can be computed as follows*

$$(\mathrm{id} + \gamma B)^{-1}(\boldsymbol{x}) = \left\{ (\bar{z}, \dots, \bar{z}) \mid \bar{z} \in (\mathrm{id} + \tfrac{\gamma}{N}G)^{-1}(\tfrac{1}{N}\sum_{i=1}^N x_i) \right\}.$$

*Proof.* Let $\boldsymbol{z} \in (\mathrm{id} + \gamma B)^{-1}(\boldsymbol{x})$. Due to the consensus constraint we have that $\boldsymbol{z} = (\bar{z}, \dots, \bar{z})$ from which the following equivalences follows:

$$
\begin{aligned}
&\boldsymbol{z} \in (\mathrm{id} + \gamma B)^{-1}(\boldsymbol{x}) \\
\iff\quad & \boldsymbol{x} \in \boldsymbol{z} + \gamma B\boldsymbol{z} \\
\iff\quad & x_i \in \bar{z} + \tfrac{\gamma}{N}G\bar{z} \quad \forall i \in [N] \\
\iff\quad & \tfrac{1}{N}\sum_{i=1}^N x_i \in \bar{z} + \tfrac{\gamma}{N}G\bar{z} \\
\iff\quad & \bar{z} \in (\mathrm{id} + \tfrac{\gamma}{N}G)^{-1}(\tfrac{1}{N}\sum_{i=1}^N x_i)
\end{aligned}
$$

This completes the proof. $\qquad\square$

With the particular choice of the operators $A, B$ in (2.4b), the coordinates of iDRS reduces into blocks. In what follows, we will refer to the variables in iDRS using bold font to make it apparent that iDRS acts on the lifted space. For convenience we will rescale the variables in iDRS by $N$ such that

$$
\begin{aligned}
\bar{\boldsymbol{x}}^k &= \tfrac{1}{N}(\bar{x}_1^k, \dots, \bar{x}_N^k), & \bar{\boldsymbol{u}}^k &= \tfrac{1}{N}(\bar{u}_1^k, \dots, \bar{u}_N^k), & \bar{\boldsymbol{y}}^k &= \tfrac{1}{N}(\bar{y}_1^k, \dots, \bar{y}_N^k) \\
\bar{\boldsymbol{s}}^k &= \tfrac{1}{N}(\bar{s}_1^k, \dots, \bar{s}_N^k), & \boldsymbol{s}^k &= \tfrac{1}{N}(s_1^k, \dots, s_N^k), & \boldsymbol{\varepsilon}_1^k &= \tfrac{1}{N}(\varepsilon_1^k, \dots, \varepsilon_N^k)
\end{aligned} \qquad (D.14)
$$

which implies that $\bar{u}_i^k \in F_i(\bar{x}_i^k)$ for all $i \in [N]$. Similarly for $\hat{\boldsymbol{p}}^k := (\mathrm{id} + \gamma B)^{-1}(\bar{\boldsymbol{x}}^k - \gamma\bar{\boldsymbol{u}}^k)$ from (D.7), it follows from Lemma D.2 that we can instead compute

$$\hat{p}^k := (\mathrm{id} + \gamma G)^{-1}\left( \tfrac{1}{N}\sum_{i=1}^N \bar{x}_i^k - \gamma\bar{u}_i^k \right)$$

for which $\hat{\boldsymbol{p}}^k = \tfrac{1}{N}(\hat{p}^k, \dots, \hat{p}^k)$.

The resulting (implicit) algorithm is specified in Algorithm III. We have used the following equivalences

$$
\begin{aligned}
\bar{u}_i^k + \bar{y}_i^k &= \gamma^{-1}(\bar{x}_i^k - \hat{p}^k) \\
\bar{s}_i^k - s_i^k &= \gamma\bar{u}_i^k + \hat{p}^k - s_i^k
\end{aligned} \qquad (D.15)
$$

since $\bar{v}_1^k = \bar{\boldsymbol{u}}^k + \bar{\boldsymbol{y}}^k$ and $\bar{v}_2^k = \hat{\boldsymbol{p}}^k - \bar{\boldsymbol{x}}^k$.

The algorithmic description in Algorithm III can explicitly be divided into an update rule for the clients and the server as described in iFedDR (Algorithm I). The adaptive stepsize $\alpha_k$ in iFedDR can be implemented in a computationally and storage efficient manner as commented on in Remark D.3.

**Remark D.3.** Evaluation of $\xi_k$, $\zeta_k$ and $\mu_k$ in (3.4) can be performed in place without the need for memory allocation of order $Nn$ as follows:

$$\xi_k = \sum_{i=1}^{N} \|\bar{x}_i^k - \hat{p}^k\|^2 = \sum_{i=1}^{N} \|\bar{x}_i^k\|^2 - 2\langle \sum_{i=1}^{N} \bar{x}_i^k, \hat{p}^k \rangle + N\|\hat{p}^k\|^2,$$

$$\mu_k = \sum_{i=1}^{N} \langle \bar{x}_i^k - \hat{p}^k, v_i^k - \hat{p}^k \rangle$$

$$= N\|\hat{p}^k\|^2 - \langle \hat{p}^k, \sum_{i=1}^{N} v_i^k \rangle - \langle \sum_{i=1}^{N} \bar{x}_i^k, \hat{p}^k \rangle + \sum_{i=1}^{N} \langle \bar{x}_i^k, v_i^k \rangle,$$

$$\zeta_k = \frac{1}{\gamma^2} \sum_{i=1}^{N} \|\bar{s}_i^k - s_i^k\|^2 = \frac{1}{\gamma^2} \sum_{i=1}^{N} \|\hat{p}^k - v_i^k\|^2$$

$$= \frac{1}{\gamma^2} \left( \sum_{i=1}^{N} \|v_i^k\|^2 - 2\langle \sum_{i=1}^{N} v_i^k, \hat{p}^k \rangle + N\|\hat{p}^k\|^2 \right),$$

with $v_i^k = s_i^k - \gamma F_i(\bar{x}_i^k)$. Note that the computation of $\zeta_k$ can be ignored entirely if the error condition in Step I.3 is relaxed slightly by only involving $\xi_k$. $\qquad\square$

---

**Algorithm III** Inexact FedDR (implicit)

---

REQUIRE $\quad s^0 \in \mathbb{R}^{Nn}$ $\lambda \in (0, 2)$, $\gamma \in (0, \infty)$, $\sigma \in (0, 1)$
REPEAT for $k = 0, 1, \ldots$ until convergence
III.1: Find

$$\bar{x}_i^k = s_i^k - \gamma(\bar{u}_i^k + \varepsilon_i^k) \quad \text{with} \quad \bar{u}_i^k \in F_i(\bar{x}_i^k)$$

$$\bar{y}_i^k = \gamma^{-1}(\bar{x}_i^k - \gamma\bar{u}_i^k) - \gamma^{-1}(\mathrm{id} + \gamma G)^{-1}\left( \frac{1}{N} \sum_{i=1}^{N} \bar{x}_i^k - \gamma\bar{u}_i^k \right)$$

$$\bar{s}_i^k = \bar{x}_i^k - \gamma\bar{y}_i^k$$

such that

$$\sum_{i=1}^{N} \|\varepsilon_i^k\|^2 \le \sigma^2 \max \left\{ \sum_{i=1}^{N} \|\bar{u}_i^k + \bar{y}_i^k\|^2, \frac{1}{\gamma^2} \sum_{i=1}^{N} \|\bar{s}_i^k - s_i^k\|^2 \right\}$$

III.2:

$$s_i^{k+1} = s_i^k - \lambda\alpha_k(\bar{u}_i^k + \bar{y}_i^k) \quad \text{with} \quad \alpha_k = \frac{\sum_{i=1}^{N} \langle \bar{u}_i^k + \bar{y}_i^k, s_i^k - \bar{s}_i^k \rangle}{\sum_{i=1}^{N} \|\bar{u}_i^k + \bar{y}_i^k\|^2}$$

RETURN $s^{k+1}$

---

**Proof of Theorem 3.1.** The proof follows directly from Theorem D.1 with the particular choice of $A$, $B$ in (2.4b). Maximally monotonicity of $A$, $B$ follows from that of $F_i$ for all $i \in [N]$ and $G$. Single-valuedness of $A$ follows from Lipschitz continuity of $F_i$ for all $i \in [N]$.

The condition $(x^\star, y^\star) \in \mathrm{zer}\, T_{\mathrm{PD}}$ is equivalent to requiring that $x^\star$ is a solution to the primal problem (2.4) and $y^\star = Ax^\star$ due to (2.6) and $A$ being singlevalued. This completes the proof. $\qquad\square$

D.3 APPLICATION OF iFEDDR TO MINIMIZATION PROBLEMS

Assumption I for the inclusion (2.1) are satisfied under the following assumption on problem (2.2) where $F_i = \nabla f_i$ and $G = \partial g$.

**Assumption III** (Requirements for problem (2.1)).

(i) *The function $f_i : \mathbb{R}^n \to \mathbb{R}$ is convex for all $i \in [N]$.*

(ii) *The gradient of $f_i$ is $L_i$-Lipschitz continuous for all $i \in [N]$, i.e.*

$$\|\nabla f_i(x) - \nabla f_i(x')\| \le L_i \|x - x'\| \quad \forall x, x' \in \mathbb{R}^n.$$

(iii) *$g : \mathbb{R}^n \to \mathbb{R}$ is proper lsc convex.*

For convenience, Algorithm IV specializes iFedDR (Algorithm I) to the case of minimization where the operator $F_i = \nabla f_i$ and $G = \partial g$.

---

**Algorithm IV** The inexact federated Douglas-Rachford algorithm (iFedDR) for minimization

---

REQUIRE   starting point $s_i^{-1} \in \mathbb{R}^n$, $\alpha_{-1} = 0 \in \mathbb{R}$, stepsize $\gamma \in (0, \infty)$, $\lambda \in (0, 2)$, and $\sigma \in (0, 1)$

REPEAT FOR $k = 0, 1, \ldots$ until convergence

IV.1:  Each client $i \in [N] := \{1, \ldots, N\}$ computes
$$s_i^k = s_i^{k-1} + \lambda \alpha_{k-1}(\bar{x}_i^{k-1} - \hat{p}^{k-1})$$
and approximately solves
$$\bar{x}_i^k \simeq \mathrm{prox}_{\gamma f_i}(s_i^k) := \arg\min_{x \in \mathbb{R}^n} f_i(x) + \frac{1}{2\gamma}\|x - s_i^k\|^2 \qquad (D.17)$$
and sends to the server
$$(\bar{x}_i^k, \nabla f_i(\bar{x}_i^k), s_i^k).$$

IV.2:  The server computes the average $\hat{p}^k := \mathrm{prox}_{\gamma g}\left(\frac{1}{N}\sum_{i=1}^N (\bar{x}_i^k - \gamma \nabla f_i(\bar{x}_i^k))\right)$ and the scalar quantities
$$\xi_k = \sum_{i=1}^N \|\bar{x}_i^k - \hat{p}^k\|^2, \ \zeta_k = \frac{1}{\gamma^2}\sum_{i=1}^N \|\gamma \nabla f_i(\bar{x}_i^k) - s_i^k + \hat{p}^k\|^2, \ \text{and } \mu_k = \sum_{i=1}^N \langle \bar{x}_i^k - \hat{p}^k, \gamma \nabla f_i(\bar{x}_i^k) - s_i^k + \hat{p}^k \rangle.$$
$$(D.18)$$
See Remark D.3 for how to carry out the computation memory-efficiently.

IV.3:  IF  $\sum_{i=1}^N \|s_i^k - \gamma \nabla f_i(\bar{x}_i^k) - \bar{x}_i^k\|^2 \leq \sigma^2 \max\{\xi_k, \zeta_k\}$ THEN
the server sends back
$$(\hat{p}^k, \bar{\alpha}_k) \quad \text{where} \quad \alpha_k = {}^{\mu_k}/_{\xi_k}.$$

IV.4:  ELSE
request the clients to refine the approximation in (D.17) to higher accuracy.

RETURN  $\hat{p}^k$

---

The resolvent of the operator $F_i = \nabla f_i$ in (3.3) reduces to finding an approximate solution to the strongly convex problem
$$\min_{x \in \mathbb{R}^n} f_i^\gamma(x) := f_i(x) + \frac{1}{2\gamma}\|x - s_i^k\|^2. \qquad (D.16)$$
which can be solved first order iterative methods such as gradient descent on the following gradient $\nabla f_i^\gamma(x) = \nabla f_i(x) + \frac{1}{\gamma}(x - s_i^k)$.

### D.4   APPLICATION OF IFedDR TO MINIMAX PROBLEMS

We will consider the following $N$ client minimax problem
$$\min_{u \in \mathbb{R}^m} \max_{v \in \mathbb{R}^r} g(u) + \frac{1}{N}\sum_{i=1}^N f_i(u, v) - h(v) \qquad (D.19)$$
Let $n = m + r$, $x = (u, v)$ and choose $G(x) := (\partial g(u), \partial h(v))$, $F_i(x) := (\nabla_u f_i(u, v), -\nabla_v f_i(u, v))$ in iFedDR.

Assumption I used in Theorem 3.1 are satisfied under the following conditions.

**Assumption IV.** *Problem* (D.19) *satisfies*

*(i) For all $i \in [N]$, the function $f_i : \mathbb{R}^{m+r} \to \mathbb{R}$ is convex-concave, i.e. $f_i(\cdot, v)$ is convex for all $v \in \mathbb{R}^r$ and $f_i(u, \cdot)$ is concave for all $u \in \mathbb{R}^m$.*

*(ii) The operator $F_i : \mathbb{R}^{m+r} \to \mathbb{R}^{m+r}$ is $L_i$-Lipschitz continuous for all $i \in [N]$, i.e.*
$$\|F_i(x) - F_i(x')\| \leq L_i\|x - x'\| \quad \forall x, x' \in \mathbb{R}^{m+r}.$$

*(iii) $g : \mathbb{R}^m \to \mathbb{R}$ and $h : \mathbb{R}^r \to \mathbb{R}$ is proper lsc convex.*

The resolvent of the operator $F_i$ in (3.3) reduces to solving
$$\min_{u \in \mathbb{R}^m} \max_{v \in \mathbb{R}^r} f_i^\gamma(u, v) := f_i(u, v) + \frac{1}{2\gamma}\|u - s_{i,u}^k\|^2 - \frac{1}{2\gamma}\|v - s_{i,v}^k\|^2 \qquad (D.20)$$

where $s_i^k = (s_{i,u}^k, s_{i,v}^k)$. The subproblem (D.20) can be approximated by a first-order iterative solver with access to the operator

$$\nabla F_i^\gamma(x) = F_i(x) + \tfrac{1}{\gamma}(x - s_i^k).$$

Note that, when $F_i$ is monotone the operator $F_i^\gamma$ is strongly monotone.

## E  BASELINE METHODS

Given a client update rule $V_i : \mathbb{R}^n \to \mathbb{R}^n$, defines the following compositions

$$V_i^k = \underbrace{V_i \circ V_i \circ \cdots \circ V_i}_{\tau \text{ times}}. \tag{E.1}$$

Let the server update be given as

$$x^{k+1} = (1 - \lambda)x^k + \frac{\lambda}{N} \sum_{i=1}^{N} V_i^\tau(x^k). \tag{E.2}$$

To recover FedAvg take $V_i(x) := x - \eta\nabla f_i(x)$ for some $\eta > 0$. Instead, to recover FedProx take $V_i(x) := x - \eta\nabla f_i^\gamma(x)$ as defined in (3.1). It is useful to note that the Lipschitz constant for $\nabla f_i^\gamma$ is

$$L_i^\gamma := \tfrac{1}{\gamma} + L_i. \tag{E.3}$$

**FedDR**  For convenience we also include the algorithmic description of FedDR (Tran Dinh et al., 2021) in our notation in Algorithm I. To compare against our proposed iFedDR method, consider the specialized case of minimization where iFedDR (Algorithm I) reduces to Algorithm IV. Originating from the extragradient error-correction step in iPPPA, our method averages $\bar{x}_i^k - \gamma\nabla f_i(\bar{x}_i^k)$ instead of $2\bar{x}_i^k - s_i^k$ in the computation of the server average $\hat{p}^k$. This modification is crucial for allowing the relative stopping criterion for the client proximal computation also appearing in our method. Lastly, iFedDR comes with an adaptive stepsize in place of $\lambda_k$ in Algorithm I, which is also a result of the error-correction step. Only in the special case of minimization when the proximal operator can additionally be computed exactly, do the two methods coincide.

---

**Algorithm I** FedDR

---

REQUIRE  starting point $s_i^{-1} \in \mathbb{R}^n$, $\bar{x}_i^{-1} = \hat{p}^{-1} \in \mathbb{R}^n$, stepsize $\gamma \in (0, \infty)$, and $\lambda_k \in (0, 2)$

REPEAT FOR $k = 0, 1, \dots$ until convergence

  I.1:  Each client $i \in [N] := \{1, \dots, N\}$ computes
$$s_i^k = s_i^{k-1} - \lambda_{k-1}(\bar{x}_i^{k-1} - \hat{p}^{k-1})$$
    and approximately solves
$$\bar{x}_i^k \simeq \mathrm{prox}_{\gamma f_i}(s_i^k) \tag{E.4}$$
    and sends to the server
$$p_i^k := 2\bar{x}_i^k - s_i^k$$

  I.2:  The server computes and sends back the average $\hat{p}^k := \mathrm{prox}_{\gamma g}\left(\frac{1}{N}\sum_{i=1}^{N} p_i^k\right)$

RETURN  $\hat{p}^k$

---

## F  EXPERIMENTS

The gradient method, $x \mapsto x - \eta F_i^\gamma(x)$ with $\eta > 0$, is used as the local solver for sake of simplicity and consistent comparison between methods (see (3.1) concerning the objective for prox-based methods). See Appendix D.3 and Appendix D.4 for details regarding the minimization case and minimax case respectively. Not surprisingly, due to the deterministic nature of the update, we do not observe significant variation in the experiments, and thus refrain from reporting standard deviations. Experiments are carried out on a cluster of Intel(R) Xeon(R) CPU E5-2680 v3  2.50GHz, each execution using 4 cores and a maximum of 8192Mb. We provide a reference implementation of iFedDR in both Jax and PyTorch.

Figure 1 uses the same hyperparameters as Figure 2, which are specified in Table 2. The indicated number of `inner` steps refers to the number of client steps used per client on average across all $m$ server iterations. The iFedDR method uses the uninformed safe default of $\gamma = 1.0$, $\lambda = 1.0$, and $\sigma^2 = 0.99$ throughout if not otherwise specified. More fine-grained recommendations are provided in Figure 8.

**Fashion-MNIST** We additionally conduct experiments on Fashion-MNIST (Xiao et al., 2017) under extreme label shift where client $i$ only has class $i$ for all $i \in N$. We fix the server steps to $K = 500$ and optimize over the client learning rate. The prox-parameter $\gamma$ in iFedDR is not optimized, but picked to be a reasonable default of $\gamma = 1.0$. See Table 5 for a summary of the hyperparameters and Figure 7 for the experimental results.

Table 2: Hyperparameters for experiments on the vehicle/`w8a` dataset.

| Hyperparameter | FedAvg | Scaffold | FedProx | FedDR | iFedDR |
|---|---|---|---|---|---|
| $\ell_2$-regularization
Server steps $K$
Server stepsize $\lambda$ | 0.00001
2000
1 | | | | |
| Client learning rate $\eta$ | $\left\{\frac{1}{L_i}, \frac{1}{10L_i}, \frac{1}{100L_i}\right\}$ | | $\left\{\frac{1}{L_i^{\gamma}}, \frac{1}{10L_i^{\gamma}}, \frac{1}{100L_i^{\gamma}}\right\}^1$ | $\frac{1}{L_i^{\gamma}}^1$ | $\frac{1}{L_i^{\gamma}}^1$ |
| Client steps $\tau$ | $\{10, 100\}$ | $\{10, 100\}$ | $\{10, 100\}$ | $\{10, 100\}$ | $10^2$ |
| Prox-parameter $\gamma$ | - | - | 1.0 | 1.0 | 1.0 |
| Error tolerance $\sigma^2$ | - | - | - | - | 0.99 |

[1] Defined in (E.3). [2] Before refinement.

Table 3: Hyperparameters for experiments on CIFAR10 with linear probing across 20 clients. Rescaling the input image to match the pre-training sample size is avoided by relying on AdaptivePooling.

| Hyperparameter | FedAvg | Scaffold | FedProx | FedDR | iFedDR |
|---|---|---|---|---|---|
| $\ell_2$-regularization
Server steps $K$
Server stepsize $\lambda$ | 0.00001
500
1 | | | | |
| Client learning rate $\eta$ | $\left\{\frac{1}{L_i}, \frac{1}{10L_i}, \frac{1}{100L_i}\right\}$ | | $\left\{\frac{1}{L_i^{\gamma}}, \frac{1}{10L_i^{\gamma}}, \frac{1}{100L_i^{\gamma}}\right\}^1$ | $\frac{1}{L_i^{\gamma}}^1$ | $\frac{1}{L_i^{\gamma}}^1$ |
| Client steps $\tau$ | $\{10, 100\}$ | $\{10, 100\}$ | $\{10, 100\}$ | $\{10, 100\}$ | $10^2$ |
| Prox-parameter $\gamma$ | - | - | 1.0 | 1.0 | 1.0 |
| Error tolerance $\sigma^2$ | - | - | - | - | 0.99 |

[1] Defined in (E.3). [2] Before refinement.

Table 4: Hyperparameters for fair classification experiments. A gridsearch was necessary for establishing convergence of FedProx and FedDR. In particular FedDR needed 1000 client iterations.

| Hyperparameter | FedProx | FedDR | iFedDR |
|---|---|---|---|
| $\ell_2$-regularization
Server steps $K$
Server stepsize $\lambda$
Max regularization $\delta$
Client learning rate $\eta$ | 0.00001
500
1
0.1
$\{0.0001, 0.001, 0.01, 0.1\}$ | | |
| Client steps $\tau$ | $\{10, 100\}$ | $\{10, 100, 1000\}$ | $10^2$ |
| Prox-parameter $\gamma$ | 1.0 | 1.0 | 1.0 |
| Error tolerance $\sigma^2$ | - | - | 0.99 |

[1] Defined in (E.3). [2] Before refinement.

Table 5: Hyperparameters for Fashion-MNIST experiments in Figure 7. We use the cross entropy loss on a linear model with softmax and full-batch gradient descent for the client update.

| Hyperparameter | FedAvg | FedProx | FedDR | iFedDR |
|---|---|---|---|---|
| $\ell_2$-regularization | 0.00001 | | | |
| Server steps $K$ | 500 | | | |
| Server stepsize $\lambda$ | 1 | | | |
| Client learning rate $\eta$ | $\{0.5, 0.1, 0.01\}$ | $\{0.5, 0.1, 0.01\}$ | $\{0.5, 0.1, 0.01\}$ | $\{0.5, 0.1, 0.01\}$ |
| Client steps $\tau$ | $\{50, 100, 200\}$ | $\{50, 100, 200\}$ | $\{50, 100, 200\}$ | $50^{1}$ |
| Prox-parameter $\gamma$ | - | 1.0 | 1.0 | 1.0 |
| Error tolerance $\sigma^2$ | - | - | - | 0.99 |

[1] Before refinement.

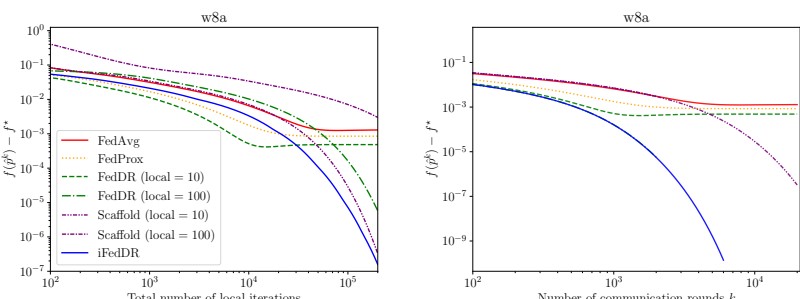

Figure 5: Logistic regression on a heterogeneous data split of `w8a`.

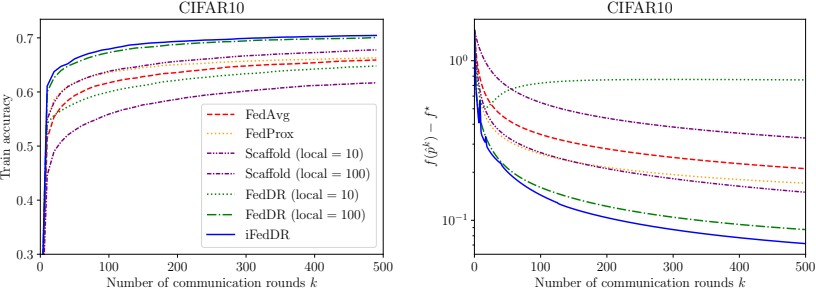

Figure 6: CIFAR10 training accuracy and optimality gap.

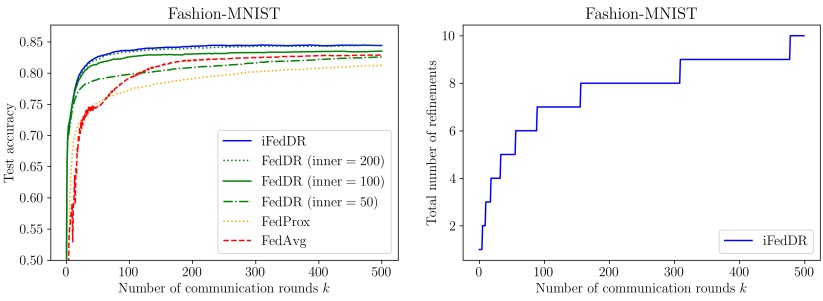

Figure 7: Fashion-MNIST under data heteogenity.

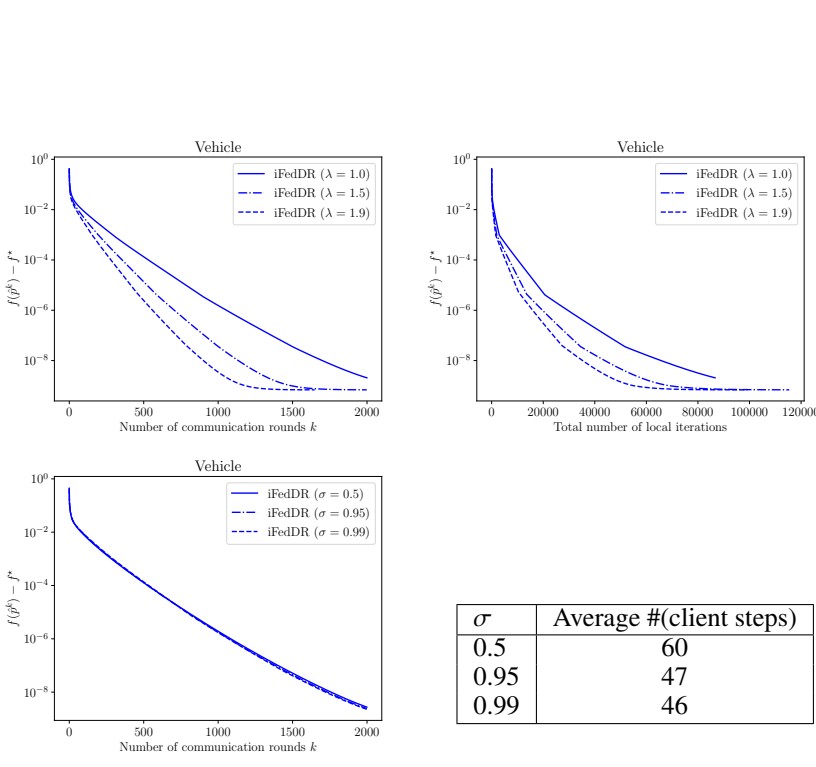

Figure 8: Hyperparameter ablation of iFedDR (default parameters are described in Table 2). (top) The theoretically optimal choice of the stepsize parameter $\lambda \in (0, 2)$ is $\lambda = 1$ for monotone problems, but especially for strongly monotone problem it is recommended to set it closer to 2 (cf. Section 1.1 regarding connection to FedSplit). (bottom) The error tolerance $\sigma \in (0, 1)$ does not have a noticeable effect on the convergence, but it changes how many client iterations are needed. It is recommended to set $\sigma$ close to 1 to minimize the number of client steps. Finally, the proximal stepsize parameter $\gamma$ can be picked arbitrarily large, but leads to a computationally harder proximal subproblem. The tradeoff is explored in Figure 1 and discussed in Remark 3.2.

