# OpenReview forum: "iFedDR: Auto-Tuning Local Computation with Inexact Douglas-Rachford Splitting in Federated Learning"
_ICLR.cc/2025/Conference — Submitted to ICLR 2025_

### Official Review · Reviewer_rCZW · 2024-11-03

**Soundness:** 3
**Presentation:** 2
**Contribution:** 3
**Rating:** 5
**Confidence:** 3

**Summary:**

This paper proposes iFedDR, which is a horizontal federated learning algorithm based on inexact Douglas-Rachford splitting. The proposed algorithm features adaptive stepsizes and an extragradient correction step. The algorithm includes an online-checkable relative error condition, which enables the server to dynamically request more accurate solutions to the local subproblems from some clients in order to guarantee convergence.

**Strengths:**

The paper is well-written with a clear flow. The theoretical results are solid and the proofs are easy to follow.

**Weaknesses:**

1. Lack of discussion of other methods for solving the primal-dual problem (2.6) (or the primal (2.4a) and the dual (2.5)). It would be helpful to include one or two sentences explaining why Douglas-Rachford splitting is chosen over other existing methods.

2. Some definition/interpretation of important variables are missing, e.g., the interpretation of $s$ in DRS and the definition of $m$ in Theorem 3.1. Please read through the main manuscript and make sure that it is self-contained.

3. The limitations of the proposed method was not discussed. A potential limitation of the proposed algorithm is its applicability in realistic federated learning tasks with deep neural network models, in which many of the assumptions (e.g., convexity) needed in the convergence analysis are violated and thus it is not clear to me if the proposed algorithm could still outperform the standard ones, e.g., FedAvg, FedSVRG, etc. I suggest the authors to add a paragraph discussing the limitations of the proposed method.

4. The numerical experiments focus on simple logistic regression and linear models. They are good starting points but are structurally very different from the high dimensional nonconvex problems in training deep networks. I suggest to authors to test the method on, for example, training full ResNet18 on CIFAR10 in the federated scenario, and to report the training efficiency and accuracy against standard methods. Without more comprehensive tests, it is difficult to assess the potential impact of the proposed method.

**Questions:**

1. The authors mention that the proposed algorithm is also applicable to constrained problems but did not discuss much further in this direction. Could the authors comment on what types of constraints (inequality, equality, etc...) can the algorithm handle? Is it simply reformulating constrained problems to the form of (2.1)? How does the number of constraints affect the dimensionality of the reformulated problem? The application to constrained problems may warrant a standalone paragraph for description and discussion, as well as perhaps some experiments to demonstrate the results.

2. How does the algorithm performs when the client loads are imbalanced (clients with different compute power, different amount of data, etc)?

3. Is the proposed method extendable to allow for asynchronous updates?

**Details Of Ethics Concerns:**

None.

---

> ### Author Response · Authors · 2024-11-20
>
> We thank the reviewer for the feedback and address all remaining concerns below.
>
> > Alternative methods to Douglas-Rachford splitting (DRS)
>
> One alternative to the primal problem (2.4a) is the forward-backward splitting (FBS) which proceeds as follows:
>
> $\mathbf x^{k+1} = (\operatorname{id} + \gamma B)^{-1}(\mathbf x^k - \gamma A(\mathbf x^k))$
>
> There are mainly two reasons for why we consider DRS instead:
>
> - FBS may not converge when the operator $A$ is merely monotone. Monotonicity is important for moving beyond minimization and capturing e.g. convex-concave minimax problems.
> - FBS applied to FL reduces to aggregating gradients on the clients and requires additional conditions on the stepsize $\gamma$. In contrast, using DRS we can move the bulk of the computation to each client (through the proximal subproblem) and take large stepsizes. Having an explicit client subproblem additionally allows us to use any fast solver for the subproblem (as comment on in l. 245-255).
>
> We thank the reviewer for the suggestion and have included a remark regarding FBS after the introduction of DRS in the updated version.
>
> > Interpretation of $s$ and definition of $m$
>
> - The integer $m$ is defined through the min-operator in Theorem 3.1(ii). Its is the total number of communication rounds, which we now explicitly state to make it more apparent.
> - The iterate $\mathbf s^k$ in DRS is the running sum of residuals and is simply an auxilliary variable. The more relevant quantities are $\mathbf u^k$ and $\mathbf v^k$ which are gauranteed to converge to a solution of the primal problem. In the context of FL, these correspond to the client states $\bar x^k_i$ and the server average $\hat p^k$ respectively.
>
> > Discussion of limitation (e.g., convexity)
>
> We have tried to be transparent about focusing on convexity (or more generally monotonicity) throughout the writeup and also explicitly mention extending to nonconvex problems as an interesting direction in the conclusion.
> With that said, it is worth mentioning that without changing our analysis, our work can immediately be generalized to all variationally stable games, which handles certain nonconvex cases even in the multiplayer setting.
>
> Regarding experiments, we go beyond other theoretical work such as the celebrated ProxSkip paper, which only tests on logistic regression on a single dataset (w8a).
> In comparison we evaluated on:
>
> - logistic regression on the vehicle dataset and w8a
> - linear probing on CIFAR10 and Fashion-MNIST
> - fair classification
>
> Evaluation on nonconvex neural network training is definitely interesting, but requires its own dedicated treatment.
>
> **Questions**:
>
> > What types of constraints can the algorithm handle?
>
> The algorithm can specifically handle _projectable_ constraints (specifically $(\operatorname{id} + \gamma G)^{-1}$ in Step I.2 of Algorithm 1 reduces to a projection, when $G=\partial g$ and $g$ is the indicator function of some closed convex set).
> A concrete example is provided in the fair classification experiments, which is a minimax problem involving a simplex constraint.
>
> > Imbalanced client loads
>
> Notice that the error condition involves a sum over the clients.
> In other words, iFedDR does not require an error condition to hold for each of the clients individually.
> So as long as the sum is small enough, some clients can have large error.
>
> > Is the proposed method extendable to allow for asynchronous updates?
>
> This is definitely an interesting direction to consider, but outside the scope of this work.

---

> > ### Comment · Reviewer_rCZW · 2024-12-02
> >
> > Thank you for addressing my comments. I will keep the rating because the rather strong assumptions (monotonicity or convexity, synchronous communication, full client participation) and limited numerical tests make it hard to assess the impact and estimate the performance of the proposed algorithm on more realistic federated learning problems.

---

> > > ### Author Response · Authors · 2024-12-02
> > >
> > > We thank the reviewer for their response.
> > >
> > > We politely disagree that the assumptions are strong:
> > >
> > > - Our analysis extends to minimax problems (and more generally $m$-player games). For this very general class of problems it is impossible to relax the assumptions to the general nonconvex case as there exist exponential lower bounds for finding local solutions even in the centralized setting [1,2]. In other words, some structure is needed when moving beyond minimization.
> > > - There are several works accepted at top tier venues like ICML and NeurIPS, that only apply to full participation under (strong) convexity minimization (more restrictive then our assumptions of monotonicity) e.g. [3,4,5].
> > >
> > > Lastly, we would like to emphasize that this is theoretical work, which introduces a method that:
> > >
> > > - allows for large stepsizes $\gamma$ which directly improves the communication rate through $\mathcal O(1/m\gamma^3)$ (see Figure 1(right) and Table 1)
> > > - automatically adjusts the local computation accordingly
> > > - extends to $m$-player games
> > >
> > > We believe these ideas are interesting for the community, which seems to be shared by Reviewer pQaJ who acknowledges the novelty of adjusting local computation and Reviewer QdPZ who states that the extensions are "non-trivial" and "important".
> > > The experiments are used to demonstrate these precise theoretical properties.
> > >
> > >
> > > [1] https://www.sciencedirect.com/science/article/pii/0885064X89900174
> > >
> > > [2] https://arxiv.org/pdf/2009.09623
> > >
> > > [3] https://arxiv.org/pdf/2202.09357
> > >
> > > [4] https://proceedings.neurips.cc/paper/2020/file/4ebd440d99504722d80de606ea8507da-Paper.pdf
> > >
> > > [5] https://openreview.net/pdf?id=W72rB0wwLVu

---

### Official Review · Reviewer_QdPZ · 2024-11-03

**Soundness:** 3
**Presentation:** 3
**Contribution:** 3
**Rating:** 6
**Confidence:** 4

**Summary:**

This paper proposes iFedDR, which can be thought of as an inexact variant of Dougals-Rachford splitting, equipped with error-correction step via extragradient computation.

**Strengths:**

- FedAvg is widely still widely used in practice, but its convergence issue in the heterogeneous setting was known, which was “fixed” by methods that employ splitting methods such as FedSplit and FedDR. The proposed method, iFedDR, extends FedDR such that inexact proximal step can be employed, which is a non-trivial and important extension.
- Since the proposed method can be more generally applied to monotone inclusion problems, the provided theory applies to minimax problems.
- The paper is generally well-written, although it is a little confusing to follow different reductions (original problem to inclusion problem, iPPA to iFedDR.)

**Weaknesses:**

- It would be great if the authors can comment on the computational aspect of the proposed algorithm. In particular, I’m a bit confused about the authors’ assertion of “computationally negligible error correction”; do you mean that $\zeta$ can be ignored in Step I.3 of Algorithm 1? In that case, does the theoretical result still apply?
- Algorithm 1 involves several matrix inversion per iteration and per client (3.3). I understand the manuscript is of theoretical nature, but some comments on the computational aspect would be helpful as the manuscript is cast as an FL algorithm.

**Questions:**

- If I understand correctly, automatically adjusting the number of location computations needed is a “trade-off” in the sense that instead the step size $\gamma$ needs to be tuned, which is used for both the client update (3.3 in Algorithm 1) and the server update (I.2 in Algorithm 1). Is there a good way to choose this $\gamma$?
- In line of above comment, could the authors comment on other federated learning works that use adaptive step size, such as [2, 3]?
- Based on Figure 1 & Remark 3.2 (v), large value of $\gamma$ leads to less number of communication steps, but increases the number of local iterations needed. Could this be alleviated by decoupling the client step size and the server step size? This would change the classical DR splitting though, but still curious.
- Remark D.3 mentions $\zeta_k$ can be ignored. Is this at least empirically tested? I believe an answer to this question is important for the authors to assert “computationally negligible” error condition, even neglecting all the matrix inversions needed in the algorithm.

[1] Mukherjee et al. (2024) Locally Adaptive Federated Learning

[2] Kim et al., (2024) Adaptive Federated Learning with Auto-Tuned Clients

---

> ### Author Response · Authors · 2024-11-20
>
> We thank the reviewer for the feedback and address all remaining concerns below.
>
> > On "computationally negligible error condition"
>
> The error condition is computationally negligible in Step I.2 of Algorithm I, since it only involves computing norms and inner products which are additionally carried out on the (computationally efficient) server.
> This is detailed in Remark D.3 referenced in the Algorithm I.
>
> Regarding $\zeta$, it is important to clarify that the error condition, $err_k \leq \sigma^2\max \{\xi_k, \zeta_k\}$, is computationally negligible even when $\zeta$ is involved. However, if a simpler error condition is desired, $\zeta$ can simply be ignored by instead requiring $err_k \leq \sigma^2\xi_k$, while the same theoretical results still apply. The stopping criterion would just become slightly more stringent.
>
> > Computation of the client update (3.3)
>
> We detail this in the paragraph "client subproblem" (l. 200-208).
> The approximate client update in (3.3) is equivalent to approximately solving a strongly monotone and Lipschitz subproblem.
> This can be done with an iterative solver such as the gradient method.
>
> In the special case of minimization this reduces to computing an approximate proximal update, i.e. approximately solving
>
> $\min_x f_i(x) + \frac{1}{2\gamma} \Vert x - s_i^k\Vert^2$
>
> We have added a reference in "client subproblem" to the Appendix D.3 which describes this special case.
>
> **Questions**:
>
> > "If I understand correctly, automatically adjusting the number of location computations needed is a “trade-off” in the sense that instead the step size $\gamma$ needs to be tuned"
>
> This proximal stepsize $\gamma$ already appears in existing prox-based methods such as FedProx and FedDR.
> So iFedDR does not introduce another $\gamma$ parameter to be tuned, but rather mitigates the need to predefine the number of local steps $\tau$.
> Note, that the choice $\gamma=1$ works well throughout our experiments.
> In the updated revision we have moved our method comparison table from the appendix to the main body (now Table 1), which hopefully makes the comparison clearer.
>
> > "In line of above comment, could the authors comment on other federated learning works that use adaptive step size, such as [1,2]?"
>
> The two works can be summarized as follows:
>
> - The reference [1] uses FedAvg with the (adaptive) Polyak stepsize as the client stepsize.
> - [2] uses an adaptive client stepsize that tries to estimate the local smoothness.
>
> Our work is orthogonal to both works, and can in fact be combined with any fast client update, including schemes with various forms of adaptive stepsize.
> This modularity is the benefit of our method having an explicit client subproblem (see Section 3, l. 245-255).
>
> Our work is not concerned with adapting the client stepsize, but rather:
>
> - The adaptive stepsize $\alpha_k$ in iFedDR, which corresponds to the _server_ stepsize in FedAvg.
> - Our main interest is in automating the _number_ of steps taken by the clients, and not auto-tuning the client stepsize.
>
>
> > Can the server step size and client step size be decoupled?
>
> The stepsizes are already decoupled in the following sense:
>
> - There is $\gamma$ which is the proximal stepsize (this controls how far the client updates can move)
> - There is the client stepsize $\eta$ used for the solver of the proximal subproblem, which is picked based on $\gamma$ through the conditioning number (see Table 1)
> - There is $\lambda$ which plays the role of the server stepsize and this can be picked independently of $\gamma$ and $\eta$
>
> > Can $\zeta_k$ be ignored?
>
> The error condition can ignore $\zeta_k$ while still maintaining the same theoretically gaurantees.
> The only consequence is that the resulting error condition is slightly more stringent (so more iterations are needed of the client subsolver).
> This is a consequence of $err_k \leq \sigma^2\max \{ \xi_k, \zeta_k \}$ being directly implied by the stricter error condition $err_k \leq \sigma^2\xi_k$.
>
>
> > Clarification of reductions
>
> Regarding the problem reformulation, the main steps carried out in Section 2 are:
>
> - We start with the original finite-sum problem (2.1)
> - We consider the lifted consensus reformulation (2.4a) which introducing a consensus constraint (this allows us to model $N$ distinct client states)
> - This is the problem we solve with an extension of Douglas-Rachford splitting
>
> To obtain iFedDR from iPPPA simply pick $T=T_{\text{PD}}$ and $P$ as detailed in (4.2).
> With the consensus constraint in (2.4b) the algorithm reduces to iFedDR.
> After making those two choices, all that remains is a sequence of algebraic steps to simplify the update expression (the step by step reduction can be found in Appendix D.1-D.2).

---

> > ### Comment · Reviewer_QdPZ · 2024-12-03
> >
> > I appreciate the authors' detailed response. I just have a minor comment: regarding $\zeta_k$, if the consequence is that *more* iterations are needed by the client subsolver, it is misleading to say $\zeta_k$ can be ignored. I hope the authors clarify this point for the final version. I have already recommended acceptance, and I keep my score.

---

> > > ### Author Response · Authors · 2024-12-03
> > >
> > > We thank the reviewer for engaging with the rebuttal and for the suggestion. We will clarify in the final version that $\zeta_k$ can be ignored in the sense that the convergence guarantee is still maintained, but that this can lead to a more stringent error condition in the specific case where $\zeta_k > \xi_k$.

---

### Official Review · Reviewer_pQaJ · 2024-11-06

**Soundness:** 3
**Presentation:** 3
**Contribution:** 3
**Rating:** 6
**Confidence:** 4

**Summary:**

The paper proposes a way to automatically adjust the local computation while preserving the convergence guarantees. In particular, it focuses on a new algorithm, iFedDR, which is an inexact version of the Douglas-Rachford approach (FedDR), which is proposed as a correct way to employ the proximal operator in the FL setting. In particular, iFedDR is an extension of previous work that incorporates an adaptive stepsize and an extragradient correction step to allow for the relaxed condition of relative inexactness. The paper focuses on providing convergence guarantees of iFedDR and tests the performance of the new approach in experiments.

**Strengths:**

The paper is well-written, and the main contributions are clear. To the best of my knowledge, this is one of the first papers to aim to automate the number of local updates needed when clients use proximal updates.

I also like the fact that the analysis was done from the monotone inclusion setup that allows direct extensions of the ideas to min-max problems and m-player games. The convergence guarantees are proven for a much more general method, iPPPA, which might be interesting in its own right.

The statement of the theorems is as expected, and from a pass on the proofs in the appendix, they look correct. I did not carefully check all the details but all steps are presented clearly and make the read of them easy.

**Weaknesses:**

I believe the paper would be improved if a table with a comparison with closely related works is presented. How the final convergence guarantees of the paper are related to the existing results? Can the main theorems capture previous convergence analysis as a special case?

I find the split of sections a bit confusing. There is Theorem 3.1 in section 3, but then later, there is a full section 5 devoted to Convergence analysis. Can the authors comment on that? Why not simply have all theorems related to convergence guarantees in one section?
In my opinion, Theorem 5.1 could be the only theorem in the main paper. The theorem 3.1 is only a corollary that can be easily not mentioned anywhere in section 3 (related to the design of the method) without affecting the flow of the paper.

On plots: The figures in the paper look like screenshots rather than a higher-quality PDF. This makes the labels and names on the axis hard to read (they are blurred).

**Questions:**

See the Weaknesses section.

---

> ### Author Response · Authors · 2024-11-20
>
> We thank the reviewer for the feedback and address all remaining concerns below.
>
> > Table for comparison
>
> The appendix already included a table comparing iFedDR with existing methods.
> We agree with the reviewer that it might be helpful to make this table more visible, so we have pulled this into the main body (see Table 1).
>
> To summarize the table, iFedDR achieves a $\mathcal O(\tfrac{1}{m \gamma^3})$ rate, where $m$ is the total number of communication rounds and $\gamma$ is the proximal stepsize.
> Notice that the rate improves with increasing $\gamma$ and that this stepsize is not restricted by the Lipschitz constant $L$. The tradeoff is that increasing $\gamma$ leads to harder client subproblems, for which the number of steps $\tau$ of the iterative solver is automatically picked.
> In comparison:
> - FedDR requires $\gamma <1/L$ in the analysis (and needs $\tau$ predefined).
> - FedProx requires $\gamma \rightarrow 0$, which in turn hurts the complexity (and needs $\tau$ predefined).
> - Scaffold obtains a $\tilde{\mathcal{O}}(L/m)$ rate, so has no parameter $\gamma$ to improve the rate (and needs $\tau$ predefined).
>
> > Section structure
>
> The main reason for leading with a section on iFedDR (including theorems) is for accessibility:
>
> - The separation allows us to discuss the structure of the algorithm in the context of FL (e.g. the client subproblem and the computation of the adaptive quantities)
> - Including the convergence result (Thm. 3.1) allows us to remark on important FL concepts like e.g. the communication rate, heterogeneity agnostic, the tradeoff between communication and local computation, etc (see Remark 3.2).
>
> Theorem 5.1 regarding iPPPA is indeed more general, but the aim with having a separate section on iFedDR is to present iFedDR and its properties without requiring understanding of the higher abstraction level of iPPPA.
> Additionally, iPPPA has its own dedicated section, since the method is interesting in its own right, and a contribution on its own. For this reason and for clarifying how we arrived at iFedDR we prefer to present both theorems.
>
> > Making the plots PDFs
>
> Thank you for the suggestion, we have updated all plots to PDFs. They are hopefully easier to read now.

---

### Meta-Review · Area_Chair_1xG5 · 2024-12-13

**Metareview:**

The paper proposes iFedDR -- a horizontal federated learning algorithm based on inexact Douglas-Rachford splitting. The authors have addressed some of the concerns but the reviewer did not satisfy with the strong assumptions that make it hard to assess the impact and estimate the performance of the proposed algorithm on more realistic federated learning problems. The reviewers think that the contributions of the paper are not enough to get accepted at this conference. Please try to relax the assumptions to improve the theory and consider other publication venues.

**Additional Comments On Reviewer Discussion:**

Strong assumptions.

---

### Decision · Program_Chairs · 2025-01-22

Reject